# Flow Matching with General Discrete Paths: A Kinetic-Optimal Perspective

**Neta Shaul**[1,†], **Itai Gat**[2], **Marton Havasi**[2], **Daniel Severo**[2], **Anuroop Sriram**[2],
**Peter Holderrieth**[3,†], **Brian Karrer**[2], **Yaron Lipman**[2], **Ricky T. Q. Chen**[2]
[1]Weizmann Institute of Science, [2]Meta FAIR, [3]MIT CSAIL
[†]Work done during internship at Meta FAIR

## Abstract

The design space of discrete-space diffusion or flow generative models are significantly less well-understood than their continuous-space counterparts, with many works focusing only on a simple masked construction. In this work, we aim to take a holistic approach to the construction of discrete generative models based on continuous-time Markov chains, and for the first time, allow the use of arbitrary discrete probability paths, or colloquially, corruption processes. Through the lens of optimizing the symmetric kinetic energy, we propose velocity formulas that can be applied to any given probability path, completely decoupling the probability and velocity, and giving the user the freedom to specify any desirable probability path based on expert knowledge specific to the data domain. Furthermore, we find that a special construction of mixture probability paths optimizes the symmetric kinetic energy for the discrete case. We empirically validate the usefulness of this new design space across multiple modalities: text generation, inorganic material generation, and image generation. We find that we can outperform the mask construction even in text with kinetic-optimal mixture paths, while we can make use of domain-specific constructions of the probability path over the visual domain.

## 1 Introduction

Generative models over discrete spaces have not seen as much progress on the methodology side compared to continuous-space counterparts. For the most part, applications such as large language modeling rely solely on autoregressive models (Radford et al., 2019; Bommasani et al., 2021). The simplicity of autoregressive modeling has also motivated people to use them for multimodal generation, where other modalities, such as images and videos, are tokenized and modeled within an autoregressive framework (Van den Oord et al., 2016; Team, 2024; Sun et al., 2024). While obtaining reasonable results, they have not yet reached the performance of continuous-space generative models such as denoising diffusion (Ho et al., 2020; Song et al., 2021) and Flow Matching models (Lipman et al., 2022; Albergo et al., 2023) for the visual-audio domains (Rombach et al., 2022; Dai et al., 2023; Esser et al., 2024; Zhou et al., 2024), where it is believed that the ability to perform iterative refinement brings significant gains (Saharia et al., 2022; Zhang et al., 2024).

A promising framework that brings iterative refinement to the discrete case is to consider the use of Markov chains within a dynamical generative framework. Many discrete-space generative flow and diffusion models have seen success in the generation of text (Austin et al., 2021; Lou et al., 2024; Shi et al., 2024; Sahoo et al., 2024; Gat et al., 2024), proteins (Campbell et al., 2024), images (Austin et al., 2021; Shi et al., 2024), and even executable code (Gat et al., 2024). However, the design space of these models is currently rather limited, with many recent works instead focusing solely on the case of masking as a corruption process (Shi et al., 2024; Sahoo et al., 2024). The masked construction is an extension of masked pretraining (Devlin, 2018; Yang, 2019), but it does not fully embody the concept of iterative refinement as it is equivalent to learning autoregressive models for every ordering (Hoogeboom et al., 2021; Chang et al., 2022), and it has been noticed that some of the recent reported progress was actually misleading due to low-precision sampling (Zheng et al., 2024) rather than the explicit design choice of masking as a corruption process. In spite of this, the masked construction has often been found to be the best performing choice out

of the limited family of corruption processes previously considered tractable (Austin et al., 2021; Campbell et al., 2024; Gat et al., 2024).

We instead take a holistic view on constructing discrete Flow Matching models, massively expanding the design space to enable arbitrary probability paths, or colloquially, corruption processes, grounded in the framework of continuous-time Markov chains (CTMC). We list our contributions:

1. Analogous to the continuous setting, we find that an infinite number of velocities can generate any given probability path. In order to reduce this search space, we consider a decomposition into a *probability-advancing* velocity and a *probability-preserving* velocity.

2. To explore the space of probability-advancing velocities, we motivate a family of closed-form velocities that be formulated as optimizing kinetic energy. In particular, we are the first to formulate velocities that can work with any choice of probability path, completely opening up the design space of probability paths, *e.g.*, domain-specific constructions, while recovering the velocities used by prior works for existing paths in the literature.

3. We also find that the probability path itself can also be optimized with the same kinetic energy criterion. A closed-form solution surprisingly recovers the mixture paths considered by Gat et al. (2024) but with novel source-dependent schedulers.

4. We derive the ELBO for discrete Flow Matching models in full generality. This leads to an improved ELBO for training mixture probability paths that has not been used before, and recovers the ELBO derived by Shi et al. (2024) for the masked construction. We find that with this ELBO, our kinetic-optimal mixture paths outperform the masked construction.

## 2 BACKGROUND: DISCRETE FLOW MATCHING

We are interested in learning a generative model that approximates a data distribution $q(x)$, where $x = (x^1, x^2, \ldots, x^D) \in \mathcal{S} = \mathcal{T}^D$ with $\mathcal{T} = [K] \triangleq \{1, 2, \ldots, K\}$ being a discrete set of possible token values, and $D \in \mathbb{N}$ is number of discrete variables. For brevity and without loss of generality, we consider all dimensions to have the same number of discrete values.

**Probability paths.** We denote by $p(x)$ and $q(x)$ the source and target, respectively, probability mass functions (PMFs) over the state space $\mathcal{S}$. We consider *probability paths* $p_t(x)$, $t \in [0, 1]$, to be time-dependent PMFs taking the form

$$p_t(x) \triangleq \sum_{x_1 \in \mathcal{S}} p_t(x|x_1) q(x_1), \text{ where } p_t(x|x_1) \triangleq \prod_{i=1}^{D} p_t(x^i|x_1^i), \qquad (1)$$

and $p_t(x^i|x_1^i)$ is a *conditional probability path* which interpolates between a simple PMF at time $t = 0$ and a delta PMF centered around $x_1^i$ at $t = 1$. That is, we assume the boundary conditions $p_0(x^i|x_1^i) = p(x^i)$ and $p_1(x^i|x_1^i) = \delta_{x_1^i}(x^i)$. Hence we can interpret these probability paths $p_t(x)$ in equation 1 as interpolating between a factorized source distribution $p(x) \triangleq \prod_{i=1}^{D} p(x^i)$ and the data distribution $q(x)$. A common family of probability paths used in previous works is the collection of *mixture paths* (Gat et al., 2024), with $x_1^i$-dependent schedulers similar to Shi et al. (2024):

$$p_t(x^i|x_1^i) = (1 - \kappa_t(x_1^i))p(x^i) + \kappa_t(x_1^i)\delta_{x_1^i}(x^i), \qquad (2)$$

where $\kappa_0(\cdot) = 0$ and $\kappa_1(\cdot) = 1$ to satisfy the boundary conditions. Specifically, with $p(x^i) = \delta_{\mathrm{m}}(x^i)$ we recover the masked construction (Shi et al., 2024; Sahoo et al., 2024).

**Probability velocities.** As our generative process, we simulate a Continuous Time Markov Chain (CTMC) $(X_t)_{t \in [0,1]}$ in $\mathcal{S}$ such that its time marginals follow a prescribed probability path,

$$X_t \sim p_t. \qquad (3)$$

In order to do so, we define the concept of a *probability velocity*, also known as a rate matrix. We say that a probability velocity $u_t$ *generates* $p_t$ if $u_t$ characterizes a Markov process $X_t$ with marginal $p_t$ (equation 3) for all $t \in [0, 1)$ in the following sense:

$$\mathbb{P}(X_{t+h} = x \mid X_t = z) = \delta_z(x) + hu_t(x, z) + o(h), \qquad (4)$$

where $o(h)$ denotes a function which is asymptotically smaller than $h$, *i.e.*, $\lim_{h \to 0} o(h)/h = 0$. Intuitively, $u_t$ describes the Markov transition of $X_t$ for small step sizes $h > 0$. We note that for equation 4 to be a valid PMF, $u_t$ must at least satisfy the *Rate Conditions*:

$$u_t(x, z) \geq 0 \text{ for all } x \neq z \text{ and } \sum_x u_t(x, z) = 0 \qquad \blacktriangleright \text{ Rate Conditions} \qquad (5)$$

**Single-variable-change probability velocities.** It is natural to consider modeling a CTMC process $X_t$ over $\mathcal{S}$ by defining a $u_t(x, z)$ for all pairs $x, z \in \mathcal{S}$. However, the state space is of size $|\mathcal{T}|^D$ so this is generally prohibitive for high dimensions. A remedy is to consider rates that only allow a state to change in a *single* variable (Campbell et al., 2022), *e.g.*, in the following example we only change the variable at the $i$-th coordinate:

$$(z^1, \ldots, z^{i-1}, \boldsymbol{z^i}, z^{i+1}, \ldots, z^D) \to (z^1, \ldots, z^{i-1}, \boldsymbol{x^i}, z^{i+1}, \ldots, z^D). \tag{6}$$

To model only such changes we restrict our attention to velocities of the form $u_t^i(x^i, z)$ that describe the probability rate between the state $z$ and the state with the $i$-th coordinate replaced, *i.e.*, as described in the r.h.s. in equation 6. We can express the full velocity $u_t(x, z)$ via $u_t^i(x^i, z)$ as

$$u_t(x, z) = \sum_{i=1}^{D} u_t^i(x^i, z) \prod_{j \neq i} \delta_{z^j}(x^j), \tag{7}$$

which states the probability velocity between two states $z \to x$ is zero if they differ by more than one variable and equal $u_t^i(x^i, z)$ if they differ by exactly one variable. Plugging this velocity into equation 4, it can be shown that (Gat et al., 2024):

$$\mathbb{P}(X_{t+h} = x \mid X_t = z) = \prod_{i=1}^{D} \left[ \delta_{z^i}(x^i) + h u_t^i(x^i, z) \right] + o(h) \tag{8}$$

This implies we can sample each variable $X_{t+h}^i$ *independently* from the distribution $\delta_{z^i}(x^i) + h u_t^i(x^i, z)$, and only incur an error of $o(h)$.

**The marginal velocity.** Previous works (Campbell et al., 2024; Gat et al., 2024) have shown that constructing a generating velocity for $p_t(x)$ can be achieved by considering only the conditional probability paths in equation 1. That is, assume we have conditional velocities $u_t^i(x^i, z^i | x_1^i)$, which are velocities in the state space $\mathcal{T}$, that generate the conditional paths $p_t(x^i | x_1^i)$ in equation 1. Then a *marginal velocity* $u_t^i(x^i, z)$ that generates $p_t(x)$ takes the form:

$$u_t^i(x^i, z) = \sum_{x_1^i \in \mathcal{T}} u_t^i(x^i, z^i | x_1^i) p_{1|t}^i(x_1^i | z) \tag{9}$$

where $p_{1|t}^i(x_1^i | z)$ is the posterior probability of the $i$-th token taking the value $x_1^i$, *i.e.*,

$$p_{1|t}^i(x^i | z) = \sum_{x_1 \in \mathcal{S}} \delta_{x_1^i}(x^i) \frac{p_t(z | x_1) q(x_1)}{p_t(z)}. \tag{10}$$

Parameterizing the *factorized posterior* $\prod_{i=1}^{D} p_{1|t}^i$ is an approach taken by prior works (Austin et al., 2021; Campbell et al., 2022). To train, a simple option is the cross-entropy objective:

$$\mathcal{L}_{\text{CE}}(\theta) = \mathbb{E}_{t \sim U[0,1], x_1 \sim q(\cdot), x \sim p_t(\cdot | x_1)} \left[ -\sum_{i=1}^{D} \log p_{1|t}^{\theta, i}(x_1^i | x) \right]. \tag{11}$$

We use this training loss for general probability paths as it is generally applicable. However, for the case of mixture paths (equation 2) it is possible to derive a tractable ELBO as the marginal $u_t$ can be written in closed form without a summation as in equation 9. We cover this later in Section 6.

## 3 SAMPLE GENERATION THROUGH THE FACTORIZED POSTERIOR

The most direct approach to sample from this model is to use the marginal velocity $u_t(x^i, z)$, *e.g.*, with a first-order sampling scheme defined by removing the $o(h)$ term in equation 8, *i.e.*, given $X_t$, we advance time with step size $h$ by sampling $X_{t+h}^i$ according to

$$X_{t+h}^i \sim \delta_{X_t^i}(\cdot) + h u_t^i(\cdot, X_t), \tag{12}$$

for each $i \in [D]$, where $u_t^i$ is computed with equation 9. However, for general discrete paths this sampling procedure is intractable for large discrete spaces $\mathcal{T}$ as computing $u_t^i(x^i, z)$ with equation 9 for all $x^i \in \mathcal{T}$ has a computational complexity of $|\mathcal{T}|^2$.

Alternatively, we propose a more efficient sampling scheme by noticing that

$$\delta_{z^i}(x^i) + h u_t^i(x^i, z) \overset{(9)}{=} \sum_{x_1^i \in \mathcal{T}} \left[ \delta_{z^i}(x^i) + h u_t^i(x^i, z^i | x_1^i) \right] p_{1|t}^i(x_1^i, z), \tag{13}$$

which leads to a sampling process that avoids computing the full marginal velocity: given the current state $X_t$, sample $X_1$ from the factorized posterior, then sample $X_{t+h}$. That is, for each $i \in [D]$,

1) Sample $X_1^i \sim p_{1|t}^i(\cdot|X_t)$; and

2) Sample $X_{t+h}^i \sim \delta_{X_t^i}(\cdot) + h u_t^i(\cdot, X_t^i|X_1^i)$.

This sampling procedure still results in $X_t$ with the same time marginals while avoiding the computational cost of the summation in equation 9. To enable the use of any step size $h$, we use a slightly modified step 2; see Appendix A for more details and pseudocode in Algorithm 1.

## 4  KINETIC OPTIMAL VELOCITIES AND PROBABILITY PATHS

We first decouple the design space of probability paths and their generating velocities, providing the means to effectively explore this large design space. This section covers two contributions: (*i*) we propose a family of kinetic optimal (KO) velocities that generates any given probability path, and (*ii*) we solve for kinetic optimal probability paths, recovering a special case of mixture paths. The first contribution enables us to work with general discrete probability paths. The second contribution justifies the choice of mixture probability paths used by Gat et al. (2024) but offers novel $x_1^i$-dependent schedulers. For both, we center our designs based on optimizing a discrete notion of kinetic energy (Peyré et al., 2019).

**Notation.** As the discussion in this section applies to arbitrary probability paths and discrete state spaces, we will use a simplified notation, where our state space is now $\mathcal{T}$ and for states we use $x, z \in \mathcal{T}$, abusing a bit the previous notation (where $x^i, z^i \in \mathcal{T}$). Furthermore, we will denote by $p_t(x)$ and $u_t(x, z)$ an arbitrary probability path and velocity field in $\mathcal{T}$, respectively.

**Continuity Equation.** Given a probability path $p_t(x)$, the entire collection of velocities $u_t(x, z)$ generating $p_t(x)$ are the solutions to the Continuity Equation (a.k.a. the Kolmogorov forward equation) that also satisfy the Rate Conditions. It is useful to formulate the Continuity Equation through the *flux* $j_t$, that is

$$\dot{p}_t(x) + \mathrm{div}_x(j_t) = 0, \qquad \forall x \in \mathcal{T}, \qquad \text{with } j_t(x, z) = u_t(x, z)p_t(z). \tag{14}$$

Intuitively, the flux $j_t(x, z)$ quantifies the amount of probability mass per unit of time moving from state $z$ to state $x$. The *divergence operator* then measures the total outgoing flux minus the total incoming flux, which in the discrete case takes the form

$$\mathrm{div}_x(j_t) = \sum_{z \neq x} j_t(z, x) - \sum_{z \neq x} j_t(x, z). \tag{15}$$

**Velocity from flux.** Given a flux $j_t$ satisfying the Continuity Equation (equation 14) we can get a velocity from the flux by defining for $x \neq z$,

$$u_t(x, z) = j_t(x, z)/p_t(z) \quad \text{if } p_t(z) > 0, \qquad \text{else } u_t(x, z) = 0, \tag{16}$$

and the case $x = z$ is uniquely set by the Rate Conditions (5), $u_t(z, z) = -\sum_{x \neq z} u_t(x, z)$. The velocity defined in this way will satisfy the Continuity Equation and the Rate Conditions if the flux satisfies the following conditions:

$$j_t(x, z) \geq 0, \text{ for } x \neq z \qquad \blacktriangleright \text{ Non-negativity} \tag{17}$$
$$p_t(z) = 0 \implies j_t(x, z) = 0 \qquad \blacktriangleright \text{ Safe Flux Condition} \tag{18}$$

Intuitively, the Safe Flux Condition ensures no flux is leaving a zero probability state $z$.

**Proposition 4.1.** *Given a non-negative safe flux $j_t$ that satisfies the Continuity Equation, the velocity defined in equation 16 satisfies the Rate Conditions and generates the $p_t$ probability path.*

**Kinetic optimality.** Motivated by the approach employed in the continuous case of minimizing the kinetic energy for the conditional velocities (Lipman et al., 2022; Shaul et al., 2023), we take a similar approach for finding velocities for the discrete case. The standard convex formulation of the kinetic energy adapted to the discrete case is (Peyré et al., 2019):

$$\min_{p_t, j_t} \quad \int_0^1 \sum_{x \neq z} \frac{w_t(x, z)}{p_t(z)} j_t(x, z)^2 dt \qquad \blacktriangleright \text{ Kinetic Energy} \tag{19a}$$
$$\text{s.t.} \quad \mathrm{div}_x(j_t) = -\dot{p}_t(x), \qquad \forall x \in \mathcal{T} \qquad \blacktriangleright \text{ Continuity Equation} \tag{19b}$$
$$j_t(x, z) \geq 0, \qquad \forall x \neq z \in \mathcal{T} \qquad \blacktriangleright \text{ Non-negative flux} \tag{19c}$$
$$p_0 = p, \quad p_1 = q \qquad \blacktriangleright \text{ Boundary conditions} \tag{19d}$$

where $w_t(x, z) > 0$ is some problem-dependent weighting; a higher weight implies a smaller flux from $z \to x$, *i.e.*, the higher this value the smaller the velocity $u_t(x, z)$. The optimality criterion (equation 19a) is the kinetic energy, equivalently $\frac{j_t(x,z)^2}{p_t(z)} = u_t(x, z)^2 p_t(z)$. The benefit of formulating in terms of the flux (instead of the velocity) is that the problem becomes convex in its unknowns $(p_t, j_t)$, and in particular the Continuity Equation constraint in (19b) is linear. Lastly, in case of $\frac{w_t(x,z)}{p_t(z)} = \infty$ the energy in equation 19a is *defined* to be 0 if $j_t(x, z) = 0$, and $\infty$ if $j_t(x, z) > 0$. Therefore, to ensures the solution $j_t^\star$ is safe (equation 18) we ask that $w_t$ satisfies:

$$p_t(z) = 0 \Rightarrow \tfrac{w_t(x,z)}{p_t(z)} = \infty \qquad \blacktriangleright \text{ Safe Weight Condition} \tag{20}$$

and that problem 19 is *feasible*, *i.e.*, it has a finite energy solution. Although problem 19 is convex, solving it for a general $w_t$ requires numerical approximation. Since we want to solve it for conditional probability paths with different $x_1 \in \mathcal{T}$, *i.e.*, $q(x) = \delta_{x_1}(x)$, this can be computationally challenging. Instead, we will explore cases of $w_t$ where problem (19) is solvable in *closed-form*. We start with assuming $p_t$ is known/given, and find the kinetic optimal velocity $u_t^\star$, then afterwards we discuss optimizing the $p_t$ as well.

## 4.1 KINETIC OPTIMAL VELOCITY

Assuming $p_t > 0$ is fixed in (19), our goal is to find the kinetic optimal solution $j_t^\star$, and consequently obtaining a velocity $u_t^\star$ via (16). One observation we make is that (19) can be efficiently solved when *symmetric*, *i.e.*, when $\frac{w_t(x,z)}{p_t(z)} = \frac{w_t(z,x)}{p_t(x)}$. As we prove in Appendix B, (19) can be efficiently solved via the following linear relaxation:

$$\sum_z \tfrac{p_t(z)}{w_t(x,z)} \left[ f_t(x) - f_t(z) \right] = \dot{p}_t(x), \qquad \forall x \in \mathcal{T} \tag{21}$$

where $f_t : \mathcal{T} \to \mathbb{R}$ is the unknown function over the state space. The linear equation in (21) is of *Laplacian form*, and many properties (including closed-form solutions) are known in many cases (Vishnoi, 2012). The solution $f_t$ to (21) is unique up to a global constant and using $f_t$ we construct the kinetic optimal flux,

$$j_t^\star(x, z) \triangleq \tfrac{p_t(z)}{w_t(x,z)} \left[ f_t(x) - f_t(z) \right]_+, \tag{22}$$

where $[s]_+ = \max \{s, 0\}$ is the ReLU operator. This provides a solution to (19) with a fixed and positive $p_t$. Consequently, using (16) we get the kinetic optimal velocity. We have shown that a certain family of kinetic optimal velocities can be computed by solving a linear system (21) for arbitrary probability paths $p_t(x)$ over state-space $\mathcal{T}$. Next we will further instantiate this family and provide some closed form solution for $j_t^\star$ and $u_t^\star$.

**Closed-form $u_t$.** We will consider the case where $w_t(x, z) = \frac{p_t(z)}{\tau_t(x)\tau_t(z)}$, and $\tau_t : \mathcal{T} \to \mathbb{R}_{\geq 0}$ is a design choice of our method. To ensure $w_t$ is safe (20) we require that $p_t(z) = 0$ implies $\tau_t(z) = 0$. The solution $f_t$ to (21)—which can be checked with substitution—is:

$$f_t(x) = \tfrac{1}{\sum_{s \in \mathcal{T}} \tau_t(s)} \tfrac{\dot{p}_t(x)}{\tau_t(x)}. \tag{23}$$

One choice is $\tau_t(x) = \mathbb{1}_{[p_t(x)>0]}$, that leads to the Kinetic Optimal flux

$$j_t^\star(x, z) = \tfrac{1}{|\mathcal{T}|} \left[ \partial_t p_t(x) - \partial_t p_t(z) \right]_+, \qquad \text{for } x \neq z \tag{24}$$

which upon converting to velocity via (16) recovers the velocity proposed in Campbell et al. (2024) for positive paths, $p_t > 0$. Note however, that the above flux is not safe (does not satisfy equation 18) and if $p_t(z) = \epsilon$ the flux $j_t^\star(x, z)$ for some general $x$ is not necessarily small, showing a potential numerical issue. Campbell et al. (2024) formulate a limit case for general $p_t$ that also requires adding an extra assumption on $p_t$ (that $p_t(x) = 0 \Rightarrow \dot{p}_t(x) = 0$), which does not hold even for common probability paths that are typically used, such as the masked mixture path with linear schedulers.

Alternatively, we propose a more numerically stable choice. Consider $\tau_t(x) = p_t(x)$, *i.e.*,

$$w_t(x, z) = 1/p_t(x). \tag{25}$$

This results in $f_t(x) = \dot{p}_t(x)/p_t(x)$, and the kinetic optimal flux in this case is:

$$j_t^\star(x, z) = [p_t(z)\dot{p}_t(x) - \dot{p}_t(z)p_t(x)]_+, \qquad \text{for } x \neq z \tag{26}$$

Note that in contrast to before, this flux is safe (satisfies equation 18) and therefore works for general $p_t$. Furthermore, (26) exhibits stable limiting behavior for continuously differentiable $p_t$: when $p_t(z) \to 0$, so too will $j^\star(x, z) \to 0$.

We note that for uniform and mask source distributions with the mixture path (2), the velocity considered by Campbell et al. (2024) and our velocity resulting from (26) coincide. However, for mixture paths (2) and general discrete paths, they generally do not coincide. Additionally, the choice of velocity in (26) also recovers the velocities used by Gat et al. (2024) for mixture probability paths. See Appendix C.1 for detailed derivations. Finally, we discuss a broader family of closed-form velocities involving different choices of $\tau_t$ in Appendix C.3, which we find can significantly boost performance at low-cost sampling regimes.

**Metric-induced** $p_t(x)$**.** The velocity resulting from (26) can be applied to any user-defined $p_t$. We propose metric-induced conditional probability paths of the form

$$p_t(x|x_1) = \text{softmax}\left(-\beta_t \mathrm{d}(x, x_1)\right), \tag{27}$$

where $\beta : [0, 1] \to \mathbb{R}_{\geq 0}$ is a monotonic scheduler with $\beta_0 = 0$, $\beta_1 = \infty$, and $\mathrm{d} : \mathcal{T} \times \mathcal{T} \to \mathbb{R}_{\geq 0}$ such that $\mathrm{d}(x, x_1) = 0 \Leftrightarrow x = x_1$, interpreted loosely as a metric over discrete values. If we apply the flux in (26) for the paths in (27) and simplify, we obtain the velocity:

$$u_t^\star(x, z|x_1) = p_t(x|x_1)[\partial_t \log p_t(x|x_1) - \partial_t \log p_t(z|x_1)]_+ \tag{28}$$
$$= p_t(x|x_1)\dot{\beta}_t[\mathrm{d}(z, x_1) - \mathrm{d}(x, x_1)]_+. \tag{29}$$

This velocity has the property that we only move from state $z$ to state $x$ if $x$ is closer than $z$ to $x_1$, *i.e.*, $\mathrm{d}(x, x_1) < \mathrm{d}(z, x_1)$, hence resulting in a flow that only moves closer to $x_1$.

## 4.2 KINETIC OPTIMAL PROBABILITY PATHS

Interestingly, for the weighting choice that we have motivated for the numerically stable velocity (25), it is also possible to solve for the kinetic optimal probability path $p_t^\star$. As we show in Appendix B, in this case, the problem (19) can be formulated equivalently as

$$\min_{a_t} \quad \int_0^1 \sum_x \dot{a}_t(x)^2 \, dt \qquad\qquad \blacktriangleright \text{ Kinetic Energy} \tag{30a}$$

$$\text{s.t.} \quad \sum_x a_t(x)^2 = 1, \qquad \forall t \in [0, 1] \qquad \blacktriangleright \text{ Hypersphere constraints} \tag{30b}$$

$$a_0(x) = \sqrt{p(x)}, \quad a_1(x) = \sqrt{q(x)} \qquad \blacktriangleright \text{ Boundary conditions} \tag{30c}$$

where $a_t(x) = \sqrt{p_t(x)}$. Problem (30) is the kinetic energy of a curve over the hypersphere connecting $\sqrt{p}$ and $\sqrt{q}$. The optimal solution thus corresponds to the geodesic curve on the hypersphere,

$$a_t(x) = \frac{\sin(1-t)\Omega}{\sin \Omega}\sqrt{p(x)} + \frac{\sin t\Omega}{\sin \Omega}\sqrt{q(x)}, \quad \text{where } \Omega = \arccos\left(\sum_z \sqrt{p(z)q(z)}\right), \tag{31}$$

and consequently the optimal probability path and velocity for (30) are

$$p_t^\star(x) = a_t^2(x), \qquad u_t^\star(x, z) = a_t^2(x)\left[\partial_t \log a_t^2(x) - \partial_t \log a_t^2(z)\right]_+ \tag{32}$$

In the particular case of conditional probability paths $q(x) = \delta_{x_1}(x)$, we get that the optimal solution recovers the *mixture path* (equation 2) with a specific $x_1$-dependent scheduler:

$$\kappa_t(x_1) = 1 - \frac{\sin^2(1-t)\Omega(x_1)}{\sin^2\Omega(x_1)}, \qquad \text{where } \Omega(x_1) = \arccos\sqrt{p(x_1)}. \tag{33}$$

This justifies the mixture paths (2) as kinetic optimal, and furthermore, it naturally utilizes an $x_1$-dependent scheduler for general source distributions $p$ when $p(x_1) > 0$.

## 5 PROBABILITY-PRESERVING VELOCITIES

While we have found a particular flux $j_t^\star$, the space of fluxes for a given $p_t$ is much larger, and in this section we show how to explore it further. We first observe that since the Continuity Equation (14) is a linear equation, any flux $j_t$ satisfying this equation can be written as a sum of two fluxes:

$$j_t = j_t^\star + j_t^\perp, \qquad \text{where } \operatorname{div}_x(j_t^\perp) = 0, \tag{34}$$

where $j_t^\star$ is a particular solution to the Continuity Equation and $j_t^\perp$ is a solution to the homogenous version of the equation, *i.e.*, *divergence-free*. We call the velocity resulting from $j_t^\perp$ a *probability-preserving*, or corrector, velocity as sampling with this velocity has $p_t$ as a steady-state. For simplicity, we mainly consider the special case of symmetric flux. Symmetry is a sufficient condition for being divergence-free as is evident from (15). A natural choice for a symmetric flux is to consider a symmetrization of (22) taking the form

$$j_t^\perp(x, z) = \tfrac{p_t(z)}{w_t(x,z)} |f_t(x) - f_t(z)|, \qquad \text{and } u_t^\perp(x, z) = j_t^\perp(x, z)/p_t(z), \tag{35}$$

for any function $f_t$. For convenience, we will simply re-use the same $f_t$ that comes from optimizing the kinetic energy (19), *e.g.* the same as in (26). In contrast to the kinetic optimal velocity, which results in a unidirectional flow in the sense that samples will only move from lower $f_t(\cdot)$ to higher $f_t(\cdot)$, the symmetric flux in (35) results in a bidirectional flow that allows equal movement between any two states with non-equal $f_t(\cdot)$. Hence $j_t^\perp$ acts as a corrector to redirect samples back to previous states in a way that leaves $p_t$ invariant.

## 6 ELBO FOR DISCRETE FLOW MATCHING

We show in Appendix D that we can produce a continuous-time ELBO bound on the likelihood $\log p_1^\theta(x_1)$ for any conditional probability path and conditional probability velocity in terms of the marginal $u_t^i(x^i, z)$ and conditional $u_t^i(x^i, z^i | x_1^i)$ as follows

$$\log p_1(x_1) \geq \int_0^1 \mathbb{E}_{x_t \sim p_t(\cdot | x_1)} \sum_{i=1}^D \Big[ u_t^i(x_t^i, x_t) - u_t^i(x_t^i, x_t^i | x_1^i) \\ + \sum_{y^i \neq x_t^i} u_t^i(y^i, x_t^i | x_1^i) \log \Big( \tfrac{u_t^i(y^i, x_t)}{u_t^i(y^i, x_t^i | x_1^i)} \Big) \Big] \mathrm{d}t \tag{36}$$

Evaluating this ELBO is difficult for the same reason as sampling in Section 3, for large discrete spaces $\mathcal{T}$ computing (9) for all $x^i \in \mathcal{T}$ has a computational complexity of $|\mathcal{T}|^2$. However, for mixture paths (2), our conditional velocity resulting from (26) is used to obtain a closed-form expression for the marginal velocity (see Appendix C.2), yielding a tractable ELBO for mixture paths:

$$\log p_1^\theta(x_1) \geq \int_0^1 \mathbb{E}_{x_t \sim p_t(\cdot | x_1)} \sum_{i=1}^D \Big[ \lambda(x_t^i) p_{1|t}^\theta(x_t^i | x_t) - \sum_{y^i} \lambda(y^i) p_{1|t}^\theta(y^i | x_t) + \\ + (1 - \delta_{x_1^i}(x_t^i)) \lambda(x_1^i) \Big( 1 + \log p_{1|t}^\theta(x_1^i | x_t) \Big) \Big] \mathrm{d}t, \tag{37}$$

where $\lambda(x) = \tfrac{\dot\kappa_t(x)}{1 - \kappa_t(x)}$. This ELBO has not been used previously, *e.g.*, Campbell et al. (2022) had to resort to a doubly stochastic estimator. Specifically for the masked construction, we recover the ELBO used by Zheng et al. (2024) for $x_1^i$-independent schedulers and used by Shi et al. (2024) for $x_1^i$-dependent schedulers; see Appendix D.1.

## 7 RELATED WORK

**Generative modeling through marginalization.** Denoising diffusion models (Sohl-Dickstein et al., 2015; Ho et al., 2020; Song et al., 2021) construct generative models by reversing a noising process. The Flow Matching framework (Lipman et al., 2022; Albergo et al., 2023; Liu et al., 2022) shares similar traits but instead constructs generative models through a marginalization of conditional Markov processes, allowing a larger design space of probability paths. These types of models can be trained at scale both efficiently and stably relative to other frameworks, and thus have seen massive success in the large-scale generation of images (Rombach et al., 2022; Esser et al., 2024), videos (Singer et al., 2022), and audio (Le et al., 2024; Vyas et al., 2023).

Figure 1: Generative perplexity vs. ELBO of kinetic optimal (KO) and linear schedulers of FineWeb-Edu models. The ELBO is evaluated: WikiText-103, LAMBADA, Penn TreeBank, FineWeb-Edu, and OpenWebText. **Bold** highlights the Pareto front.

Table 1: Zero-shot unconditional perplexity bound as in equation 36 of Fineweb-Edu models, more details are in Appendix E.1. $^*$ denotes our reimplementation of the method.

| METHOD | LAMBADA↓ | WIKITEXT2↓ | PTB↓ | WIKITEXT103↓ | 1BW↓ | OPENWEBTEXT↓ | FINEWEB-EDU (TRAIN SET)↓ |
|---|---|---|---|---|---|---|---|
| SEDD$^*$ (mask) (Lou et al., 2024) | ≤58.57 | ≤42.84 | ≤ 136.99 | ≤42.88 | ≤114.17 | ≤36.55 | ≤19.41 |
| MD4$^*$ (Shi et al., 2024) | ≤61.27 | ≤43.08 | ≤157.00 | ≤43.02 | ≤127.55 | ≤35.57 | ≤18.69 |
| DFM - Linear ($\beta_0 = 1024$) | ≤60.59 | ≤44.17 | ≤180.75 | ≤44.29 | ≤147.21 | ≤36.33 | ≤18.67 |
| DFM - Kinetic Optimal (mask) | ≤58.5 | ≤41.80 | ≤144.46 | ≤41.83 | ≤123.83 | ≤35.57 | ≤18.71 |
| DFM - Kinetic Optimal ($\beta_0 = 1024$) | ≤58.41 | ≤42.19 | ≤147.09 | ≤42.34 | ≤115.51 | ≤36.07 | ≤18.63 |

**Continuous-time Markov Chains.** These aforementioned frameworks have been adapted to the discrete domain by making use of Continuous-time Markov Chains (CTMC) as the choice of generative process (Campbell et al., 2022; 2024; Lou et al., 2024; Gat et al., 2024). Many works discuss both a uniform noise and mask construction (Campbell et al., 2024; Lou et al., 2024); however, more recent works have focused more and more on the simple masked construction where each discrete variable is randomly replaced with a dummy or mask token (Sahoo et al., 2024; Shi et al., 2024) as it often performs favorably compared to uniform noise. However, the simple masked construction leads to a generative model that is equivalent to any-order autoregressive modeling under mild assumptions (Hoogeboom et al., 2021; Zheng et al., 2024).

**Any-order autoregressive modeling.** While autoregressive models prespecify a fixed ordering, any-order autoregressive models learn conditional probabilities for every ordering. Training is often carried out by randomly masking out parts of the data sample (Devlin, 2018; Yang, 2019). Some works have focused on architectural choices: Germain et al. (2015) randomly masks out weights to induce a randomized ordering, while Pannatier et al. (2024) uses a fixed causal attention architecture but randomly permutes the input ordering, with the end goal of learning all combinations of conditional distributions so that generation of the variables can be done in any order. The ordering itself is often optimized further by the use of heuristic scoring functions (Chang et al., 2022; Ziv et al., 2024).

## 8 EXPERIMENTS

We evaluate Discrete Flow Matching (DFM) on multiple modalities: text, crystalline material, and image generation. Our main goal is to show that Discrete Flow Matching can outperform autoregressive models, and within the class of Discrete Flow Matching, we explore new additions such as the kinetic optimal and the metric-induced constructions. In text, we mainly explore the kinetic optimal probability paths (equation 33) with different source distributions, as these have access to the closed-form ELBO (37). In material generation, we find that enabling permutation invariance for DFM easily outperforms autoregressive models at de novo generation, achieving state-of-the-art results. Furthermore, in domains where a natural metric exists, we demonstrate our method's ability to inject inductive bias into the velocity and probability path using equations 27 and 29. We show that our large design space enables competitive results even with non-mask probability paths, showcasing the capabilities of our expanded design space.

### 8.1 TEXT GENERATION

We explore our method on the task of text generation. We use the kinetic optimal probability path as in equation 33, which only has one hyper-parameter, the source distribution $p(x)$. For source distribution, we compute the statistics of tokens appearances in the training data $p_{\text{stats}}(x^i)$ and construct a single-parameter family of source distributions:

$$p(x) = \prod_{i=1}^{D} p(x^i), \quad p(x^i) = \text{softmax}(-\beta_0 \log p_{\text{stats}}(x^i)), \tag{38}$$

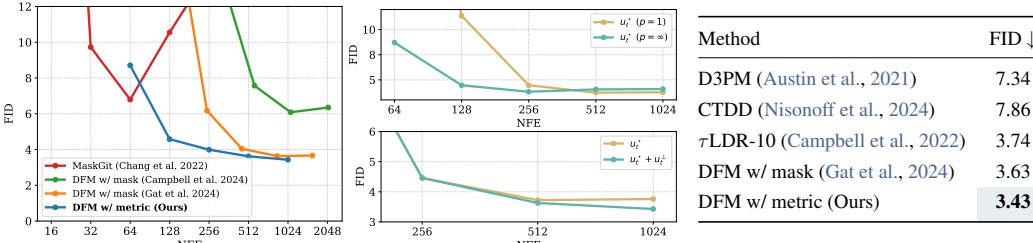

Figure 2: (*left*) Increasing the design space of discrete probability paths and velocities allows us to perform better than prior works, while significantly boosting performance at the low NFE regime. (*middle*) We find that the choice of kinetic optimal $u_t^\star$ significantly affects the low NFE regime while adding the probability-preserving component $u_t^\perp$ stabilizes the high NFE regime. (*right*) Comparison of FID values for discrete generative models.

Table 2: De novo material generation. Our primary metric, *Stability Rate*, is the fraction of materials with energies below the convex hull formed by stable materials, following Miller et al. (2024).

| Method | NFE | Validity (%) ↑ | | Coverage (%) ↑ | | Property ↓ | | Stability Rate (%) ↑ |
|---|---|---|---|---|---|---|---|---|
| | | Structural | Composition | Recall | Precision | wdist ($\rho$) | wdist ($N_{el}$) | |
| CDVAE (Xie et al., 2021) | 5000 | 100.00 | 86.70 | 99.15 | 99.49 | 0.688 | 0.278 | 1.57 |
| DiffCSP (Jiao et al., 2023) | 1000 | 100.00 | 83.25 | 99.71 | 99.76 | 0.350 | 0.125 | 5.06 |
| FlowMM (Miller et al., 2024) | 1000 | 96.85 | 83.19 | 99.49 | 99.58 | 0.239 | 0.083 | 4.65 |
| CrystalLLM (70B) (Gruver et al., 2024) | – | 99.6 | 95.4 | 85.8 | 98.9 | 0.81 | 0.44 | 5.28 |
| Autoregressive | – | 86.43 | 89.33 | 63.31 | 99.74 | 0.088 | 0.030 | 1.99 |
| Perm. invariant DFM - Mask w/ Cubic | 250 | 94.40 | 84.40 | 98.25 | 99.40 | 0.244 | 0.144 | 6.90 |
| Perm. invariant DFM - Mask w/ Kinetic Optimal | 250 | 95.79 | 88.50 | 90.11 | 99.29 | 0.542 | 0.154 | **7.02** |

where $\beta_0 = -1$ recovers the data statistics, $\beta_0 = 0$ yield a uniform distribution on all tokens. Also, $\beta_0 \to \infty$ yields a uniform distribution on the set of least probable tokens in the data, which behaves similarly to a mask source distribution.

For this experiment, we used linear and kinetic optimal schedulers with mask, $p(x) = \delta_{\mathrm{m}}(x)$, and $\beta_0 \in \{-0.5, 0.0, 0.5, 1, 2, 4, 64, 256, 1024\}$ source distributions. The models are trained on the FineWeb-Edu (Lozhkov et al., 2024) data set. Table 1 compares the evidence lower bound (ELBO), as in equation 37, of our trained models with previous works. See Appendix E.1 for experimental setup. We find that the kinetic optimal scheduler yields the best results on most of the evaluation sets. Notably, to the best of our knowledge, this is the first time a non-mask source distribution obtains comparable results and sometimes outperforms the mask source distribution. Figure 1 presents a view of the generative perplexity (as measured by GPT2-large) vs. the ELBO of each model. Generative perplexity represents the likelihood as determined by an external model, whereas the ELBO indicates the likelihood as assessed by the evaluated model. We see that models trained using the kinetic optimal scheduler achieve better tradeoffs than those trained with the linear scheduler, as they more frequently appear on the Pareto front.

## 8.2 CRYSTALLINE MATERIAL GENERATION

To showcase the flexibility of our approach, we use discrete Flow Matching to generate crystals. We train on inorganic materials from the MP-20 dataset, a subset of the Materials Project database (Jain et al., 2013). Crystalline materials are represented using a combination of continuous and discrete variables, which we tokenize using the same method as Gruver et al. (2024), which fine-tunes a 70B LlaMa-2 autoregressive model (Touvron et al., 2023). In contrast, we are the first to perform crystal generation with a purely discrete non-autoregressive model.

An important distinction is that since discrete Flow Matching directly predicts the factorized posterior (10), we can easily impose permutation invariance of the atoms, which should significantly reduce the complexity of the learning problem. This is as opposed to prior works on using autoregressive models for material generation (Flam-Shepherd & Aspuru-Guzik, 2023; Gruver et al., 2024) which must impose an unnatural ordering on the variables. We show results in Table 2 where we achieve state-of-the-art results using discrete Flow Matching, in particular, with a kinetic optimal scheduler (33). We believe non-autoregressive generation is a key ingredient in performing well due to the ability to impose structure such as permutation invariance. Compared to continuous-space

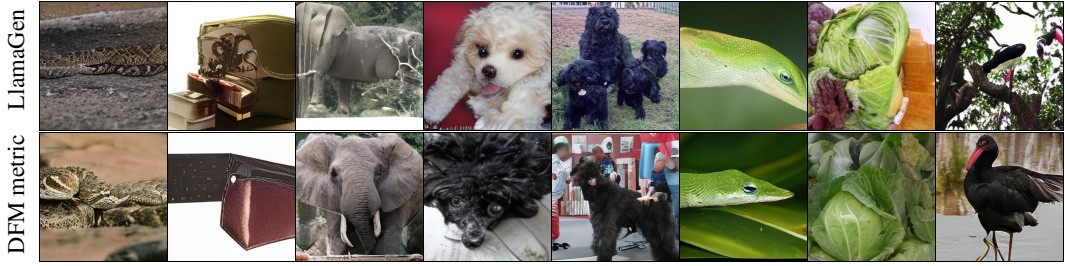

Figure 3: Generated samples for ImageNet 256×256, with the same class label per column. (*top*) Autoregressive LlamaGen model (Sun et al., 2024). (*bottom*) Discrete Flow Matching with metric-induced probability path (27).

models such as FlowMM (Miller et al., 2024) and DiffCSP (Jiao et al., 2023), we see a large performance gain in terms of our main metric, stability rate ($\geq 38\%$ relative improvement), from using discrete generative models due to the discrete nature of crystal generation.

## 8.3 PIXEL SPACE IMAGE GENERATION

We first consider the case of image generation in pixel space. Here, $\mathcal{T} = \{0, \ldots, 255\}$ and we have access to a natural choice of metric, by embedding $\mathcal{T}$ on the interval $[-1, 1] \subset \mathbb{R}$ and using the Euclidean distance $d(x, y) = |x - y|$ in (27), as is typically done for continuous-space image generative models. We use the CIFAR-10 dataset (Krizhevsky et al., 2009) for these experiments. Results are shown in Figure 2, where we can improve upon the masked construction while also retaining performance at low number of function evaluations (NFE). Generated samples are shown in Figure 3 and in Appendix G. We find that optimizing the velocity after training can provide significant gains: the choice of probability-advancing velocity (Appendix C.3) affects the low NFE samples while the adding the probability-preserving component (Section 5) improves at high NFE.

## 8.4 DISCRETE LATENT IMAGE GENERATION

We also explore the use of discrete Flow Matching as a generative model within a discrete latent space learned by a vector quantized variational autoencoder (VQVAE; Van Den Oord et al. (2017)). We use images from face-blurred ImageNet (Deng et al., 2009; Chrabaszcz et al., 2017) at 256×256 resolution. For training the VQVAE model, we follow the setup in Sun et al. (2024) and use 16× downsampling to produce a latent space of dimen-

Table 3: Face-blurred ImageNet-256 with the Llama-B architecture (111M parameters). * denotes our reimplementation.

| Method | NFE | FID |
|---|---|---|
| LlamaGen (AR) (Sun et al., 2024) | 256 | 5.46 |
| LlamaGen (AR)* (Sun et al., 2024) | 256 | 4.81 |
| DFM - Mask* (Gat et al., 2024) | 100 | 5.72 |
| DFM - Metric (Ours) | 100 | **4.50** |

sion 16×16 with a codebook size of $|\mathcal{T}| = 2^{14} = 16384$. As our choice of $d(\cdot, x_1)$, we use the same metric that was used to train the VQVAE model, which is $d(x, y) = \|x/\|x\| - y/\|y\|\|$. We show quantitative results in Table 3, where we find that discrete Flow Matching model with the metric probability path outperforms the autoregressive approach, while the masked construction lags behind. In addition, we show generated samples in Figure 3 and Figure 8, along with a visualization of the metric probability path in Figure 4 and ablation studies on NFE and CFG scale in Appendix G.

## 9 CONCLUSION

We have opened up the design space of discrete Flow Matching models based on the the continuous-time Markov chain generative process. In particular, we propose a kinetic optimal point of view for constructing velocities given prescribed probability paths. This leads to, for the first time, allowing arbitrary probability paths to be used. Furthermore, we justify mixture paths with particular schedulers as being kinetic optimal solutions, and showcase for the first time, competitive results for non-mask source distributions. Our method naturally encapsulates existing approaches, and we showcase the flexibility of our approach to designing discrete Flow Matching models across multiple application domains, ranging from text generation, to materials and image generation, where we see significant gains over autoregressive models.

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

# A  ALWAYS-VALID SAMPLING SCHEME

The second step of the sampling scheme defined in Section 3 requires the condition $h \leq \frac{1}{\left|u_t(z^i, z^i, x_1^i)\right|}$ to be a valid PMF, allowing only small step sizes. To avoid this constraint on $h$, we use an alternative first-order sampling scheme. We replace step 2 from Section 3 with

2) Sample $X_{t+h}^i \sim e^{-h\lambda(X_t^i|X_1^i)} \delta_{X_t^i}(\cdot) + (1 - e^{-h\lambda(X_t^i|X_1^i)}) \frac{u_t(\cdot, X_t^i|X_1^i)}{\lambda(X_t^i|X_1^i)}(1 - \delta_{X_t^i}(\cdot)),$

where $\lambda(X_t^i|X_1^i) = \left|u_t(X_t^i, X_t^i|X_1^i)\right|$.

Interpreting this expression, $e^{-h\lambda(X_t^i|X_1^i)}$ is the probability the state does not change. If we do not change state, then we sample from $\delta_{X_t^i}(\cdot)$. If we do change state, then we sample from $\frac{u_t(\cdot, X_t^i|X_1^i)}{\lambda(X_t^i|X_1^i)}(1 - \delta_{X_t^i}(\cdot))$, which is a normalized distribution over all states not equal to $X_t^i$.

This is still a first-order sampling scheme, *i.e.* it is $o(h)$ error from $\mathbb{P}(X_{t+h}^i \mid X_t^i)$. However, unlike the simple Euler procedure, this alternative is always a valid PMF for any step size $h$.

---

**Algorithm 1** Euler Solver

---

**Require:** model $\theta$, $x_0$, $h$
  $t \leftarrow 0$
  $X_t \leftarrow x_0$
  **while** $t < 1$ **do**
    **for** i=0,...,D {in parallel} **do**
      $X_1^i \sim p_{1|t}^{\theta,i}(\cdot|X_t)$
      $\lambda^i \leftarrow \left|u_t^i(X_t^i, X_t^i|X_1^i)\right|$
      $Z_{\text{jump}}^i \sim U[0,1]$
      **if** $Z_{\text{jump}}^i \leq 1 - e^{-h\lambda^i}$ **then**
        $X_t^i \sim \frac{u_t^i(\cdot, X_t^i|X_1^i)}{\lambda^i}\left(1 - \delta_{X_t^i}(\cdot)\right)$
      **end if**
    **end for**
    $t \leftarrow t + h$
  **end while**
  **return** $X_t$

---

# B  SYMMETRIZED KINETIC OPTIMIZATION PROBLEM

What makes the optimization of the kinetic energy problem (19) hard for a general weighting $w_t$ is the non-negative flux constraint (19c) which is non-linear. Nevertheless, for a symmetric weighting defined as

$$\frac{w_t(x, z)}{p_t(z)} = \frac{w_t(z, x)}{p_t(x)}, \quad \forall\, x, z \in \mathcal{T}, \tag{39}$$

we are able to overcome this constraint and obtain a close-form solution of the kinetic optimal flux $j_t^\star$. Specifically, we prove that for a symmetric weighting (39) and any probability path $p_t$, the optimization of the flux is reduced to the *relaxed kinetic optimal problem*,

$$\min_{j_t} \quad \int_0^1 \sum_{x \neq z} \frac{w_t(x, z)}{p_t(z)} j_t(x, z)^2 dt \qquad \blacktriangleright \text{ Kinetic Energy} \tag{40a}$$

$$\text{s.t.} \quad \text{div}_x(j_t) = -\dot{p}_t(x), \qquad \forall x \in \mathcal{T} \qquad \blacktriangleright \text{ Continuity Equation} \tag{40b}$$

which is similar to the original problem but without the non-negativity constraint (19c).

**Lemma B.1.** *Consider a probability path $p_t > 0$ and assume a symmetric weighting $w_t$ (39), then if $j_t$ is a solution to the relaxed kinetic optimal problem (40) then*

$$j_t^\star(x, z) = [j_t(x, z) - j_t(z, x)]_+ \tag{41}$$

*is a solution to the kinetic optimal problem (19).*

*Proof of lemma B.1.* Every flux $j_t$ can be written as the sum of a symmetric matrix and anti-symmetric matrix,

$$j_t(x, z) = j_t^{\text{sym}}(x, z) + j_t^{\text{anti}}(x, z), \tag{42}$$

where

$$j_t^{\text{sym}}(x, z) = \frac{j_t(x, z) + j_t(z, x)}{2}, \quad j_t^{\text{anti}}(x, z) = \frac{j_t(x, z) - j_t(z, x)}{2}. \tag{43}$$

The divergence of any symmetric flux vanishes (15), *i.e.*,

$$\text{div}_x(j_t^{\text{sym}}) = 0. \tag{44}$$

Hence writing the kinetic optimal problem (19) for the flux in terms of the symmetric and anti-symmetric flux components gives,

$$\min_{j_t^{\text{sym}}, j_t^{\text{anti}}} \quad \int_0^1 \sum_{x \neq z} \frac{w_t(x,z)}{2p_t(z)} \left[ j_t^{\text{sym}}(x, z)^2 + j_t^{\text{anti}}(x, z)^2 \right] dt \quad \blacktriangleright \text{ Kinetic Energy} \tag{45a}$$

$$\text{s.t.} \quad \text{div}_x(j_t^{\text{anti}}) = -\dot{p}_t(x), \quad \forall x \in \mathcal{T} \quad \blacktriangleright \text{ Continuity Equation} \tag{45b}$$

$$j_t^{\text{sym}}(x, z) \geq \left| j_t^{\text{anti}}(x, z) \right|, \quad \forall x \neq z \in \mathcal{T} \quad \blacktriangleright \text{ Non-negative flux} \tag{45c}$$

Notice that for the symmetric term $j_t^{\text{sym}}$ we are left with optimization of a quadratic function that is constrained only by equation 45c which bounds it below by a non-negative value. Hence for any $j_t^{\text{anti}}$ the optimal symmetric term is

$$j_t^{\text{sym}}(x, z) = \left| j_t^{\text{anti}}(x, z) \right|, \quad \forall x, z \in \mathcal{T}. \tag{46}$$

So the solution for kinetic optimal problem is

$$j_t^\star(x, z) = j_t^{\text{anti},\star}(x, z) + \left| j_t^{\text{anti},\star}(x, z) \right| = \left[ j_t^{\text{anti},\star}(x, z) - j_t^{\text{anti},\star}(z, x) \right]_+, \tag{47}$$

where $j_t^{\text{anti},\star}(x, z)$ is an optimal anti-symmetric term. Finally, notice that the optimization problem for the anti-symmetric term *i.e.*, equations 45a and 45b, is exactly our relaxed kinetic optimal problem (40). $\qquad \square$

The relaxed kinetic optimal problem (40) is actually easy to solver and can be reduced to a linear system of equations, leading to our main proposition.

**Proposition B.2** (Kinetic-optimal relaxation)**.** *Consider a probability path $p_t > 0$ and assume a symmetric weighting $w_t$ (39). Let $f_t$ be a solution to equation 21, which is unique up to a constant. Then, $j_t^\star$ in equation 22 is the unique solution to the Kinetic Optimality problem in equation 19.*

*Proof of proposition B.2.* By lemma B.1 the kinetic optimal flux is

$$j_t^\star(x, z) = \left[ j_t^{\text{relax}}(x, z) - j_t^{\text{relax}}(z, x) \right]_+, \tag{48}$$

where $j_t^{\text{relax}}$ is the solution to the relaxed kinetic optimal problem (40). Since the continuity equation constraint (40b) is linear, we can solve the relaxed problem using Lagrange multipliers.

Denoting the Lagrange multipliers by $\lambda_t(x), \ x \in \mathcal{T}$, the solution to the relaxed kinetic optimal problem (40) is

$$j_t^{\text{relax}}(x, z) = \frac{p_t(z)}{2w_t(x, z)} \left( \lambda_t(z) - \lambda_t(x) \right), \tag{49}$$

where the Lagrange multipliers are determined by the linear system of equations,

$$\sum_{y \in \mathcal{T}} \frac{p_t(z)}{w_t(x, z)} \left( \lambda_t(x) - \lambda_t(y) \right) = \dot{p}_t(x), \quad \forall x \in \mathcal{T}. \tag{50}$$

Denoting the solution to this linear system of equation as $f_t(x)$, the optimal flux is,

$$j_t^\star(x, z) = \left[ \frac{p_t(z)}{2w_t(x, z)} \left( f_t(z) - f_t(x) \right) - \frac{p_t(x)}{2w_t(z, x)} \left( f_t(x) - f_t(z) \right) \right]_+ \tag{51}$$

$$= \frac{p_t(z)}{w_t(x, z)} \left[ f_t(z) - f_t(x) \right]_+, \tag{52}$$

where we have used the symmetric weighting assumption in the second equality. $\qquad \square$

**Proposition B.3** (Kinetic Optimal paths.). *For $p_t > 0$ and the choice of $w_t(x, z) = \frac{1}{p_t(x)}$, the solution to the Kinetic Optimality problem in equation 19 is equivalent to problem 30.*

*Proof.* According to equation 22 the optimal flux in this case takes the form $j_t^*(x, z) = p_t(x)p_t(x)\left[\frac{\dot{p}_t(x)}{p_t(x)} - \frac{\dot{p}_t(z)}{p_t(z)}\right]_+$. Plugging this in problem 19 we get the energy

$$
\begin{aligned}
\sum_{x,z} p_t(x)p_t(z)\left(\frac{\dot{p}_t(x)}{p_t(x)} - \frac{\dot{p}_t(z)}{p_t(z)}\right)_+^2 &= \frac{1}{2}\sum_{x,z} p_t(x)p_t(z)\left(\frac{\dot{p}_t(x)}{p_t(x)} - \frac{\dot{p}_t(z)}{p_t(z)}\right)^2 \\
&= \sum_x p_t(x)\left(\frac{\dot{p}_t(x)}{p_t(x)}\right)^2 - \left(\sum_x p_t(x)\frac{\dot{p}_t(x)}{p_t(x)}\right)^2 \\
&= \sum_x \left(\frac{\dot{p}_t(x)}{\sqrt{p_t(x)}}\right)^2 \\
&= 2\sum_x \left(\frac{d}{dt}\sqrt{\bar{p}_t}(x)\right)^2
\end{aligned}
$$

where in the previous to last equality we used the fact that $\sum_x \dot{p}_t(x) = \frac{d}{dt}\sum_x p_t(x) = 0$. We are left with the following optimization problem:

$$
\begin{aligned}
\min_{p_t} \quad & \int_0^1 \sum_x \left(\frac{d}{dt}\sqrt{\bar{p}_t}(x)\right)^2 & \text{(53a)} \\
& p_t(x) > 0 & \text{(53b)} \\
& \sum_x p_t(x) = 1 & \text{(53c)} \\
& p_0 = p, \quad p_1 = q & \text{(53d)}
\end{aligned}
$$

Making the change of variables $a_t(x) = \sqrt{p_t(x)}$, we get the form in equation 30, as desired. $\qquad\square$

**The case of $q = \delta_{x_1}$.** Given that $q = \delta_{x_1}$ we get that the Kinetic Optimal solution equation 32 takes a form of a mixture path (equation 2) with the scheduler specified in equation 33. Specifically we show that the probability path is

$$
p_t(x|x_1) = a_t^2(x) = \frac{\sin^2(1-t)\Omega}{\sin^2\Omega}p(x) + \left(1 - \frac{\sin^2(1-t)\Omega}{\sin^2\Omega}\right)\delta_{x_1}(x), \tag{54}
$$

and the velocity is

$$
u_t(x, z|x_1) = \frac{2\Omega}{\tan(1-t)\Omega}\left(\delta_{x_1}(x) - \delta_z(x)\right). \tag{55}
$$

We start by substituting the $q \equiv \delta_{x_1}$ in equation 31,

$$
a_t(x) = \frac{\sin(1-t)\Omega}{\sin\Omega}\sqrt{p(x)} + \frac{\sin t\Omega}{\sin\Omega}\delta_{x_1}(x). \tag{56}
$$

Hence the probability path is

$$p_t(x|x_1) = \left( \frac{\sin(1-t)\Omega}{\sin \Omega} \sqrt{p(x)} + \frac{\sin t\Omega}{\sin \Omega} \delta_{x_1}(x) \right)^2 \tag{57}$$

$$= \frac{\sin^2(1-t)\Omega}{\sin^2 \Omega} p(x) + \left( \frac{2\sqrt{p(x_1)}\sin(1-t)\Omega \sin t\Omega}{\sin^2 \Omega} + \frac{\sin^2 t\Omega}{\sin^2 \Omega} \right) \delta_{x_1}(x) \tag{58}$$

$$= \frac{\sin^2(1-t)\Omega}{\sin^2 \Omega} p(x) + \left( \frac{2\cos \Omega \sin(1-t)\Omega \sin t\Omega}{\sin^2 \Omega} + \frac{\sin^2 (\Omega - (1-t)\Omega)}{\sin^2 \Omega} \right) \delta_{x_1}(x) \tag{59}$$

$$= \frac{\sin^2(1-t)\Omega}{\sin^2 \Omega} p(x) + \left( \frac{2\cos \Omega \sin(1-t)\Omega \sin (\Omega - (1-t)\Omega)}{\sin^2 \Omega} \right. \tag{60}$$

$$\left. + \frac{(\sin \Omega \cos(1-t)\Omega - \cos \Omega \sin(1-t)\Omega)^2}{\sin^2 \Omega} \right) \delta_{x_1}(x) \tag{61}$$

$$= \frac{\sin^2(1-t)\Omega}{\sin^2 \Omega} p(x) + \left( \frac{-\cos^2 \Omega \sin^2(1-t)\Omega}{\sin^2 \Omega} + \frac{\sin^2 \Omega \cos^2(1-t)\Omega}{\sin^2 \Omega} \right) \delta_{x_1}(x) \tag{62}$$

$$= \frac{\sin^2(1-t)\Omega}{\sin^2 \Omega} p(x) + \left( \frac{-(1 - \sin^2 \Omega) \sin^2(1-t)\Omega}{\sin^2 \Omega} \right. \tag{63}$$

$$\left. + \frac{\sin^2 \Omega(1 - \sin^2(1-t)\Omega)}{\sin^2 \Omega} \right) \delta_{x_1}(x) \tag{64}$$

$$= \frac{\sin^2(1-t)\Omega}{\sin^2 \Omega} p(x) + \left( 1 - \frac{\sin^2(1-t)\Omega}{\sin^2 \Omega} \right) \delta_{x_1}(x), \tag{65}$$

where in the third equality we used $\Omega = \arccos \sqrt{p(x_1)}$. Substituting

$$\kappa_t(x_1) = 1 - \frac{\sin^2(1-t)\Omega}{\sin^2 \Omega} \tag{66}$$

in equation 75 yields the desired velocity as in equation 55.

## C   CLOSED-FORM KINETIC OPTIMAL VELOCITIES

### C.1   KINETIC OPTIMAL VELOCITIES FOR MIXTURE PATHS

We examine the velocities from the kinetic optimal fluxes given in (24) and (26) under mixture paths (2). We show (24) and (26) produce the same velocity for the uniform mixture for which $p(x) = 1/|\mathcal{T}|$, and different velocities for non-uniform mixtures. We also demonstrate our kinetic optimal velocity from (26) is the velocity proposed by Gat et al. (2024) for any mixture path.

**Positive mixture paths using (24):**   For (24), we only consider mixture paths for which $p_t(x|x_1) > 0$ for all $x \in \mathcal{T}$, including uniform $p(x) = 1/|\mathcal{T}|$ as a special case. For these mixture paths, where we recall that $x \neq z$, we have

$$u_t^\star(x,z|x_1) = \frac{[\partial_t p_t(x|x_1) - \partial_t p_t(z|x_1)]_+}{|\mathcal{T}| p_t(z|x_1)}$$

$$= \frac{\dot{\kappa}_t(x_1) [\delta_{x_1}(x) - \delta_{x_1}(z) + p(z) - p(x)]_+}{|\mathcal{T}| p_t(z|x_1)} \tag{67}$$

We now examine the uniform and arbitrary $p(x) > 0$ cases separately.

**Uniform mixture using (24):**   For uniform, we have for $x \neq z$

$$u_t^\star(x,z|x_1) = \frac{\dot{\kappa}_t(x_1) [\delta_{x_1}(x) - \delta_{x_1}(z) + p(z) - p(x)]_+}{|\mathcal{T}| p_t(z|x_1)}$$

$$= \frac{\dot{\kappa}_t(x_1) [\delta_{x_1}(x) - \delta_{x_1}(z)]_+}{|\mathcal{T}| p_t(z|x_1)} \tag{68}$$

This is only positive if $x = x_1$ and $z \neq x_1$. In which case, we have

$$
\begin{aligned}
u_t^\star(x_1, z \neq x_1 | x_1) &= \frac{\dot{\kappa}_t(x_1)\,[1-0]_+}{1 - \kappa_t(x_1)} \\
&= \frac{\dot{\kappa}_t(x_1)}{1 - \kappa_t(x_1)}
\end{aligned}
\tag{69}
$$

So in total we have

$$
u_t^*(x, z | x_1) = \frac{\dot{\kappa}_t(x_1)}{1 - \kappa_t(x_1)}(\delta_{x_1}(x) - \delta_z(x))
\tag{70}
$$

**Arbitrary $p(x) > 0$ using (24):**  For a non-uniform positive $p(x)$ we do not arrive at the same velocity as the uniform mixture. Consider $x \neq z$, $x \neq x_1$, and $z \neq x_1$, then

$$
u_t^\star(x \neq x_1, z \neq x_1 | x_1) = \frac{\dot{\kappa}_t(x_1)\,[p(z) - p(x)]_+}{|\mathcal{T}|p_t(z|x_1)}.
\tag{71}
$$

This is not zero if $p(z) > p(x)$ for any pair of $z$ and $x$, proving this is a different velocity in general.

**Arbitrary mixture paths using (26):**  Substituting in the mixture path, where we recall that $x \neq z$ and $p_t(z|x_1) > 0$, we have

$$
\begin{aligned}
u_t^\star(x, z | x_1) &= \frac{1}{p_t(z|x_1)} \left[ \partial_t p_t(x|x_1) p_t(z|x_1) - \partial_t p_t(z|x_1) p_t(x|x_1) \right]_+ \\
&= \dot{\kappa}_t(x_1) \left[ \delta_{x_1}(x) - p(x) - \frac{p_t(x|x_1)}{p_t(z|x_1)}(\delta_{x_1}(z) - p(z)) \right]_+
\end{aligned}
\tag{72}
$$

We consider several cases. First, $x \neq x_1$ and $z = x_1$, then the term in brackets is negative and hence $u_t^* = 0$. Second if $x \neq x_1$ and $z \neq x_1$, we have

$$
\begin{aligned}
u_t^\star(x \neq x_1, z \neq x_1 | x_1) &= \dot{\kappa}_t(x_1) \left[ -p(x) + \frac{p_t(x|x_1)p(z)}{p_t(z|x_1)} \right]_+ \\
&= \dot{\kappa}_t(x_1) \left[ -p(x) + \frac{(1 - \kappa_t(x_1))p(x)p(z)}{(1 - \kappa_t(x_1))p(z)} \right]_+ \\
&= 0.
\end{aligned}
\tag{73}
$$

Our final case, $x = x_1$ and $z \neq x_1$, gives

$$
\begin{aligned}
u_t^\star(x_1, z \neq x_1 | x_1) &= \dot{\kappa}_t(x_1) \left[ 1 - p(x_1) + \frac{((1 - \kappa_t(x_1))p(x_1) + \kappa_t(x_1))\,p(z)}{(1 - \kappa_t(x_1))p(z)} \right]_+ \\
&= \frac{\dot{\kappa}_t(x_1)}{1 - \kappa_t(x_1)} \left[ (1 - \kappa_t(x_1))(1 - p(x_1)) + (1 - \kappa_t(x_1))p(x_1) + \kappa_t(x_1) \right]_+ \\
&= \frac{\dot{\kappa}_t(x_1)}{1 - \kappa_t(x_1)}
\end{aligned}
\tag{74}
$$

So in total for any mixture path we have

$$
u_t^*(x, z | x_1) = \frac{\dot{\kappa}_t(x_1)}{1 - \kappa_t(x_1)}(\delta_{x_1}(x) - \delta_z(x)),
\tag{75}
$$

recovering the velocity proposed in Gat et al. (2024).

## C.2   MARGINAL VELOCITY IN CLOSED-FORM FOR MIXTURE PATHS

As shown in Appendix C.1, the kinetic optimal flux given by (26) results in kinetic optimal velocity (75) for mixture paths. To derive the marginal velocity, we insert (75) into (9) as follows

$$u_t^i(x^i, z) = \sum_{x_1^i \in \mathcal{T}} u_t(x^i, z^i | x_1^i) p_{1|t}^i(x_1^i | z)$$

$$= \sum_{x_1^i \in \mathcal{T}} \frac{\dot{\kappa}_t(x_1^i)}{1 - \kappa_t(x_1^i)} (\delta_{x_1^i}(x^i) - \delta_{z^i}(x^i)) p_{1|t}^i(x_1^i | z)$$

$$= \frac{\dot{\kappa}_t(x^i)}{1 - \kappa_t(x^i)} p_{1|t}^i(x^i | z) - \delta_{z^i}(x^i) \sum_{x_1^i \in \mathcal{T}} \frac{\dot{\kappa}_t(x_1^i)}{1 - \kappa_t(x_1^i)} p_{1|t}^i(x_1^i | z). \tag{76}$$

### C.3 Power $\infty$ velocity for general paths

We begin by defining a single parameter family of kinetic optimal velocities. For every $\alpha > 1$ the flux as in equation 22 for $\tau_t(x) = p_t^\alpha(x)$ is

$$j_t^\star(x, z) = p_t^\alpha(x) p_t^\alpha(z) \left[ f_t(x) - f_t(z) \right]_+, \quad f_t(x) = \frac{1}{\sum_{s \in \mathcal{T}} p_t^\alpha(s)} \frac{\dot{p}_t(x)}{p_t^\alpha(x)}. \tag{77}$$

Further simplifying $j_t^\star(x, z)$,

$$j_t^\star(x, z) = \left[ \dot{p}_t(x) \frac{p_t^\alpha(z)}{\sum_{s \in \mathcal{T}} p_t^\alpha(s)} - \dot{p}_t(z) \frac{p_t^\alpha(x)}{\sum_{s \in \mathcal{T}} p_t^\alpha(s)} \right]_+. \tag{78}$$

An interesting case of the flux above is taking the limit $\alpha \to \infty$, where

$$\frac{p_t^\alpha(x)}{\sum_{s \in \mathcal{T}} p_t^\alpha(s)} \xrightarrow[\alpha \to \infty]{} \delta_{\arg\max_s(p_t(s))}(x), \tag{79}$$

and the flux is

$$j_t^\star(x, z) = \left[ \dot{p}_t(x) \delta_{\arg\max_s(p_t(s))}(z) - \dot{p}_t(z) \delta_{\arg\max_s(p_t(s))}(x) \right]_+. \tag{80}$$

Indeed the above flux satisfy the Continuity Equation and the Rate Conditions as in Indeed the above flux satisfy the Continuity Equation and the Rate Conditions as in equation 17. Note that it can also be seen that

$$j_t^\star(x, z) \xrightarrow[p_t(z) \to 0]{} 0. \tag{81}$$

## D Evidence lower bound (ELBO) for CTMC

Let $0 = t_0 < t_1 < \cdots < t_K = 1$ be a uniform discretization of the interval $[0, 1]$ with $h = t_{k+1} - t_k = \frac{1}{K}$. Also let $q_{k+1|k}(x^i | z^i, x_1^i) = \delta_{z^i}(x^i) + h u_t(x^i, z^i | x_1^i)$ be the Euler discretization of the variational process, and let $p_{k+1|k}(x^i | z^i) = \delta_{z^i}(x^i) + h u_t^i(x^i, z)$ be the Euler discretization of the learned process, with both starting at the same source distribution $q_0(x^i | x_1^i) = p(x^i)$. We also assume the model $p(x_1^i | x_{0:K}^i) = \delta_{x_K^i}(x_1^i)$. The discrete-time ELBO is then

$$\log p_\theta(x_1) \geq \mathbb{E}_{x_{0:K} \sim q_{0:K}(\cdot | x_1)} \left[ \log p(x_1 | x_{0:K}) + \log p_{0:K}(x_{0:K}) - \log q_{0:K}(x_{0:K} | x_1) \right] \tag{82}$$

$$= \mathbb{E}_{x_{1:K} \sim q_{1:K}(\cdot | x_1)} \sum_{i=1}^{D} \left[ \log \delta_{x_K^i}(x_1^i) - \sum_{k=0}^{K-1} D_{\mathrm{KL}}(q_{k+1|k}(x_{k+1}^i | x_k, x_1^i) \| p_{k+1|k}(x_{k+1}^i | x_k)) \right] \tag{83}$$

$$- \sum_{i=1}^{D} \cancel{D_{\mathrm{KL}}(q_0(x^i | x_1^i) \| p(x^i))} \tag{84}$$

Each term in the summation:

$$D_{\mathrm{KL}}(q_{k+1|k}(x^i|z, x_1^i)\|p_{k+1|k}(x^i|z)) \tag{85}$$

$$= \sum_{x^i} q_{k+1|k}(x^i|z, x_1^i) \log \frac{q_{k+1|k}(x^i|z, x_1^i)}{p_{k+1|k}(x^i|z)} \tag{86}$$

$$= \sum_{x^i} \left[\delta_{z^i}(x^i) + h u_t^i(x^i, z^i|x_1^i)\right] \log \frac{\delta_{z^i}(x^i) + h u_t^i(x^i, z^i|x_1)}{\delta_{z^i}(x^i) + h u_t^i(x^i, z)} \tag{87}$$

$$= \left[1 + h u_t(z^i, z^i|x_1^i)\right] \log \frac{1 + h u_t^i(z^i, z|x_1^i)}{1 + h u_t^i(z^i, z)} + h \sum_{x^i \neq z^i} \left[u_t(x^i, z^i|x_1^i)\right] \log \frac{u_t^i(x^i, z^i|x_1^i)}{u_t^i(x^i, z)} \tag{88}$$

Taylor series expansion around $h = 0$:

$$\log(1 + h u_t^i) = h u_t^i + o(h) \tag{89}$$

So we can simplify

$$D_{\mathrm{KL}}(q_{k+1|k}(x^i|z, x_1^i)\|p_{k+1|k}(x^i|z)) \tag{90}$$

$$= \left[1 + h u_t^i(z^i, z^i|x_1^i)\right]\left(h u_t^i(z^i, z^i|x_1^i) - h u_t^i(z^i, z)\right) + h \sum_{x^i \neq z^i} \left[u_t^i(x^i, z^i|x_1^i)\right] \log \frac{u_t^i(x^i, z^i|x_1^i)}{u_t^i(x^i, z)} + o(h) \tag{91}$$

$$= h \left(u_t^i(z^i, z^i|x_1^i) - u_t^i(z^i, z) + \sum_{x^i \neq z^i} \left[u_t^i(x^i, z^i|x_1^i)\right] \log \frac{u_t^i(x^i, z^i|x_1^i)}{u_t^i(x^i, z)}\right) + o(h) \tag{92}$$

Taking limit as $K \to \infty$, hence $h = \frac{1}{K} \to 0$, and asserting that $q(x_K^i|x_1^i) = \delta_{x_1^i}(x_K^i)$ in this continuous-time limit, we obtain the ELBO:

$$\log p_\theta(x_1) \geq \tag{93}$$

$$\int_0^1 \mathbb{E}_{x_t \sim p_t(\cdot|x_1)} \sum_{i=1}^{D} \left[u_t^i(x_t^i, x_t) - u_t^i(x_t^i, x_t^i|x_1^i) + \sum_{x \neq x_t} u_t^i(x^i, x_t^i|x_1^i) \log \frac{u_t^i(x^i, x_t)}{u_t(x^i, x_t^i|x_1^i)}\right] \mathrm{d}t \tag{94}$$

### D.1 ELBO FOR MASKED MODELS

The masked probability path is as in equation 2 with source distribution $p^i(x^i) = \delta_{\mathrm{m}}(x^i)$. Assuming the model is such that $p_{1|t}^\theta(z^i|x) = \delta_{x_1^i}(z^i)$ if $x^i$ is unmasked (i.e. $x^i = x_1^i$), our ELBO as in equation 37 further simplifies to

$$\log p_1^\theta(x_1) \geq \int_0^1 \mathbb{E}_{x_t \sim p_t(\cdot|x_1)} \sum_{i=1}^{D} \delta_{\mathrm{m}}(x_t^i)\left[-\sum_{y^i} \frac{\dot{\kappa}_t(y^i)}{1 - \kappa_t(y^i)} p_{1|t}^\theta(y^i|x_t)\right. \tag{95}$$

$$\left. + \frac{\dot{\kappa}_t(x_1^i)}{1 - \kappa_t(x_1^i)}\left(1 + \log p_{1|t}^\theta(x_1^i|x_t)\right)\right] \mathrm{d}t. \tag{96}$$

This simplified expression recovers the ELBO for masked mixture path as proposed by Shi et al. (2024).

# E EXPERIMENTAL DETAILS

## E.1 TEXT GENERATION

**Data.** Our model are on trained OpenWebText (Gokaslan & Cohen, 2019) and FineWeb-Edu (Lozhkov et al., 2024). For evaluation we use the test split of five dataset Radford et al. (2019): WikiText-103, WikiText-2 Merity et al. (2016), LAMBADA Paperno et al. (2016), PennTreebank (PTB) Marcus et al. (1993), One Billion Words (1BW) Chelba et al. (2014). Additionally, we extract 512 samples of length 1024 tokens of GPT2 Tokenizer from FineWeb-Edu, we do not see on training (our models do not complete an epoch in this dataset.

**Models.** All of our text generation models uses DiT transformers architecture Peebles & Xie (2022) with 12 layers, 12 attention heads, and hidden dimension of 768 ($150m$ parameters). For optimization we use constant learning rate of $3e^{-4}$ with 2500 warmup steps, Adam optimizer with $\beta_1 = 0.9$ and $\beta_2 = 0.999$, and weight decay of 0.03. We also use a dropout rate of 0.02, and we train for $200k$ iterations with batch size of 512.

**ELBO for training.** All text model are trained using our ELBO for mixture path as in equation 37. To avoid exploding terms in the loss, we sample $t$ in $[0, 1 - 1e^{-3}]$.

**ELBO for evaluation.** We want to evaluate the ELBO as in equation 37 for trained models with the mixture path as in equation 2. We note that each choice of scheduler $\kappa_t(x_1^i)$ will results in a different conditional probability path and hence a different different ELBO. However for every token independent scheduler $\kappa_t(x_1^i) \equiv \kappa_t$ we can change the integration variable from $t$ to $\kappa$,

$$\log p_1^\theta(x_1) \geq \int_0^1 dt \mathbb{E}_{x_t \sim p_t(\cdot|x_1)} \sum_{i=1}^N \left[ \frac{\dot{\kappa}_t(x_t^i)}{1 - \kappa_t(x_t^i)} p_{1|t}^\theta(x_t^i|x_t) - \sum_{y^i} \frac{\dot{\kappa}_t(y^i)}{1 - \kappa_t(y^i)} p_{1|t}^\theta(y^i|x_t) + \right. \tag{97}$$

$$\left. + (1 - \delta_{x_1^i}(x_t^i)) \frac{\dot{\kappa}_t(x_1^i)}{1 - \kappa_t(x_1^i)} \left( 1 + \log p_{1|t}^\theta(x_1^i|x_t) \right) \right] \tag{98}$$

$$= \int_0^1 dt \mathbb{E}_{x_t \sim p_t(\cdot|x_1)} \frac{\dot{\kappa}_t}{1 - \kappa_t} \sum_{i=1}^N \left[ p_{1|t}^\theta(x_t^i|x_t) - \sum_{y^i} p_{1|t}^\theta(y^i|x_t) + \right. \tag{99}$$

$$\left. + (1 - \delta_{x_1^i}(x_t^i)) \left( 1 + \log p_{1|t}^\theta(x_1^i|x_t) \right) \right] \tag{100}$$

$$= \int_0^1 \frac{d\kappa}{1 - \kappa} \mathbb{E}_{x_t \sim p_{t_\kappa}(\cdot|x_1)} \sum_{i=1}^N \left[ p_{1|t_\kappa}^\theta(x_t^i|x_t) - \delta_{x_1^i}(x_t^i) \right. \tag{101}$$

$$\left. + (1 - \delta_{x_1^i}(x_t^i)) \left( \log p_{1|t_\kappa}^\theta(x_1^i|x_t) \right) \right], \tag{102}$$

where $t_\kappa$ is the inverse of $\kappa_t$. For token dependent schedulers we only use the Kinetic Optimal scheduler as in equation 33,

$$\kappa_t(x_1^i) = \frac{\sin^2(1 - t)\Omega(x_1^i)}{\sin^2 \Omega(x_1^i)}, \qquad \text{where } \Omega(x_1^i) = \arccos \sqrt{p(x_1^i)}. \tag{103}$$

Note that $\Omega \in \left[0, \frac{\pi}{2}\right]$, depending on $\sqrt{p(x_1^i)}$, we take $\Omega = \frac{\pi}{4}$ and evaluate the integral,

$$\log p_1^\theta(x_1) \geq \int_0^1 dt \mathbb{E}_{x_t \sim p_t(\cdot|x_1)} \sum_{i=1}^N \left[ \frac{\dot\kappa_t(x_t^i)}{1 - \kappa_t(x_t^i)} p_{1|t}^\theta(x_t^i|x_t) - \sum_{y^i} \frac{\dot\kappa_t(y^i)}{1 - \kappa_t(y^i)} p_{1|t}^\theta(y^i|x_t) + \quad (104)$$

$$+ (1 - \delta_{x_1^i}(x_t^i)) \frac{\dot\kappa_t(x_1^i)}{1 - \kappa_t(x_1^i)} \left( 1 + \log p_{1|t}^\theta(x_1^i|x_t) \right) \Bigg] \qquad (105)$$

$$\int_0^1 d\kappa \left( \Omega = \frac{\pi}{4} \right) \mathbb{E}_{x_t \sim p_t(\cdot|x_1)} \frac{1}{\dot\kappa_{t_\kappa} \left( \Omega = \frac{\pi}{4} \right)} \sum_{i=1}^N \left[ \frac{\dot\kappa_t(x_t^i)}{1 - \kappa_t(x_t^i)} p_{1|t_\kappa}^\theta(x_t^i|x_t) \right. \qquad (106)$$

$$- \sum_{y^i} \frac{\dot\kappa_t(y^i)}{1 - \kappa_t(y^i)} p_{1|t_\kappa}^\theta(y^i|x_t) + \qquad (107)$$

$$+ (1 - \delta_{x_1^i}(x_t^i)) \frac{\dot\kappa_t(x_1^i)}{1 - \kappa_t(x_1^i)} \left( 1 + \log p_{1|t_\kappa}^\theta(x_1^i|x_t) \right) \Bigg], \qquad (108)$$

where $\kappa_t\left(\Omega = \frac{\pi}{4}\right)$ is the Kinetic Optimal scheduler with $\Omega = \frac{\pi}{4}$, and $t_\kappa$ is the inverse of $\kappa_t\left(\Omega = \frac{\pi}{4}\right)$. Now that we have a more fair estimator all the schedulers we us, for each $x_1$ we discretize $\kappa \in [0, 1 - 1e^{-4}]$ to 1024 using

$$\kappa_j = (j + \epsilon) \frac{1 - 1e^{-4}}{1024}, \qquad j = 0, ..., 1023, \epsilon \sim U[0, 1]. \qquad (109)$$

### E.2 INORGANIC MATERIAL GENERATION

**Material representation.**  A crystal is represented by a parallelepiped in 3D space with periodic boundary conditions, as in previous works (Miller et al., 2024; Xie et al., 2021). The model input is a variable-length sequences with length $6 + 4 \cdot a$, where $a$ is the number of atoms in the unit cell. The first 3 tokens represent the lengths of the sides of the parallelepiped, while the next 3 represent the angles between the sides. Every atom is comprised of 4 tokens: a discrete atom type and 3 continuous numbers representing the atom position inside the parallelopiped in cartesian coordinates. The coordinates are represented relative to the side lengths of the parallelopiped, and are therefore restricted to the interval $[0, 1]$ (known as *fractional coordinates*).

While lengths, angles, and fractional coordinates are all continuous quantities, we discretize them uniformly to generate tokens, following the same tokenization method from Gruver et al. (2024) – lengths (in Å) are truncated to one decimal place, angles (in degrees) are represented as integers, and fractional coordinates are truncated to two decimal places. The token set for these attributes can be created by the following python code:

```python
tokens_lens = [f"{i/10:.1f}" for i in range(500)]
tokens_angles = [str(x) for x in range(180)]
tokens_frac = [f"0.{i:02d}" for i in range(100)] + ["1.00"]
```

Tokens for atoms are taken from Pymatgen (Ong et al., 2013) like so

```python
from pymatgen.core.periodic_table import Element
tokens_atom = [Element.from_Z(z).name for z in range(1, 95)]
```

The overall vocabulary is composed of all previously mentioned sub-vocabularies, plus 3 special tokens: beggining-of-sentence (BOS), masking, and padding, totalling $500 + 180 + 101 + 94 + 3 = 878$.

**Model implementation.**  All of our models listed in Table 2, namely, DFM, Kinetic Optimal DFM (KO-DFM), and Autoregressive (AR), use a modified version of the Diffusion Transformer (DiT) (Peebles & Xie, 2023) implementation from Lou et al. (2024).

Two sequences that differ only in a permutation of their atoms, along with their fractional coordinates, represent the same crystal. For DFM and KO-DFM, we modified DiT to account for this

invariance by transforming the input before applying the attention mechanism. We flatten each quadruple of embeddings representing an atom (i.e., atom type plus 3 fractional coordinates) and apply a linear layer with a SiLU (Elfwing et al., 2018) activation to create a single representation for the atom. This brings the sequence length from $6 + 4 \cdot a$ to $6 + a$. Positional embeddings are then added, where the same positional embedding is added to all $a$ output embeddings of the previous step, which establishes the invariance. After the attention mechanism, 4 independent linear layers are applied to each of the $a$ outputs, increasing the sequence length from $6 + a$ back to $6 + 4 \cdot a$, before computing the logits.

For the AR model, we replaced rotary embeddings (Su et al., 2024) with sinusoidal positional encodings. Note that permutation invariance cannot be enforced in the same way as DFM and KO-DFM, as the model generates tokens auto-regressively. The AR model performs conditional generation by generating an embedding for the number of atoms $a \in \{0, ..., a_{\max} - 1\}$, where $a_{\max} = 20$ for the MP-20 dataset in Table 2. The embedding is then passed to the same conditioning mechanism (adaLN) present in the original DiT architecture (Peebles & Xie, 2023).

**Training and sampling.** Hyperparameter values used during training are listed in Table 4. DFM and KO-DFM use the same values.

| Param. | Hidden dim. | Attn. Blocks | Attn. Heads | Dropout | Batch Size | Learn. rate |
|---|---|---|---|---|---|---|
| AR | 288 | 16 | 16 | 0.1 | 1024 | 1e−3 |
| DFM, KO-DFM | 256 | 16 | 16 | 0.1 | 1024 | 1e−3 |

Table 4: Hyperparameters used to train the DiT models for material generation.

The hidden dimension of KO-DFM and DFM was lowered to roughly match the same number of parameters as the AR model and FlowMM (Miller et al., 2024) (around 25 million), due to the additional layers required to ensure permutation invariance. Models are trained to predict the next token by minimizing the cross-entropy loss (equation 11).

During sampling, the softmax temperature was fixed to 0.7 for DFM and KO-DFM, and 1.0 for the AR model. Both DFM and KO-DFM have noise distribution equal to a delta function on the all masked sequence (as in Gat et al. (2024)). DFM uses the convex linear scheduler ($\kappa_t = t$), while KO-DFM uses the proposed kinetic-optimal scheduler (33).

**Evaluation metrics** Our primary metric for material generation is based on thermodynamic stability, a key indicator of the synthesizability of materials. Thermodynamic stability is measured by comparing the energy of a material to a database of previously known materials with the same elements. Formally, we define Energy above Hull ($E^{hull}$) as the distance in energy landscape between the generated material and a convex hull of energies constructed from these reference database of materials. Stable materials have $E^{hull} < 0$, that is the energy of the new material is below the convex hull. Following Miller et al. (2024), we define our *Stability Rate* metric as the percentage of generated materials that are stable, i.e. $E^{hull} < 0$ and n-ary $\geq 2$, where n-ary of a material is the number of unique elements in it.

To compute the energies, we follow the methodology from Miller et al. (2024): we first perform structure relaxations using the CHGNet model (Deng et al., 2023), followed by density functional theory (DFT) (Kohn & Sham, 1965) calculations. We generated 10,000 materials to compute the stability rate.

Due to the high computational cost of performing these energy calculations, Xie et al. (2021) proposed a number of proxy metrics, which we also include for completeness:

1. *Structural Validity*: Percentage of generated materials where all pairwise interatomic distances are greater than 0.5 Å.

2. *Compositional Validity*: Percentage of generated materials that are determined to be charge-neutral using the SMACT heuristic system Davies et al. (2019).

3. *Coverage Precision & Recall*: Precision and Recall metrics computed by comparing 10000 generated structures to the MP-20 test set. Precision is the percentage of generated structures that are close to some test structure, while recall is the percentage of test structures

which are close to some generated structure. Closeness is evaluated using structural and compositional fingerprints (Zimmermann & Jain, 2020; Ward et al., 2016).

4. *Wasserstein Distances of Property Distributions*: Wasserstein distances between the distribution of computed properties between the test set and the generated materials. We compute these distances for two properties: density ($\rho$), and number of unique atoms ($N_{\text{el}}$)

We emphasize that most of these proxy metrics have become saturated and are not very good at distinguishing state-of-the-art models.

### E.3 IMAGE GENERATION - CIFAR10

**Models**    All our CIFAR10 models use the U-Net architecture as in Dhariwal & Nichol (2021), with channels 96 , depth 5, channels multiple [3,4,4], heads channels 64, and attention resolution 16. Additionally, we make two changes to the architecture as done in Gat et al. (2024): (i) We replace the first layer with an embedding table of size $256 \times 96$, and we stack the channel features such that the input to the U-Net is of shape $288 \times 32 \times 32$. (ii) We enlarge the size of the final layer to output a tensor of shape $3 \times 32 \times 32 \times 256$. Overall parameters count of 113M. For optimization we use dropout rate of 0.3, and Adam optimizer with $\beta_1 = 0.9$ and $\beta_2 = 0.999$, a learning rate of 1e-4. We trained with an effective batch size pf 512 for approximately 300K iterations.

**The conditional path.**    For our metric induced probability path (27) on pixel space we have a natural choice of metric. We embed $\mathcal{T} = \{0, ..., 255\}$ in the interval $[-1, 1] \subset \mathbb{R}$ using the map $\text{emb}(x) = \frac{2}{255}x - 1$ and with the $l_p$ distance,

$$\text{d}(x, x_1) = |\text{emb}(x) - \text{emb}(x_1)|^{\text{lp}},$$

where lp is a Hyper-parameter. For the $\beta_t$ scheduler we use,

$$\beta_t = c \left( \frac{t}{1-t} \right)^a,$$

where $a$ and $c$ are Hyper-parameters. We find that best results are achieved with lp $= 3$, $a = 5$, and $c = 1$. For the other baselines in Figure 2 we follow (Gat et al., 2024).

### E.4 IMAGE GENERATION - FACE-BLURRED IMAGENET256×256

Our ImageNet256 experiments are conducted on the face-blurred variant of the ImageNet benchmark dataset scaled to 256x256 pixels. We first train a tokenizer model (encoder, quantizer and decoder) that maps the images to a discrete latent representation and back. Then, we train a latent generative model to generate latent representations conditional on the image class.

**Tokenizer details.**    The tokenizer is realized as a VQVAE. Our architecture matches that of VQ-GAN (Esser et al., 2021). It applies a 16x downscaling to the image with a vocabulary size of 16384. The VQVAE is trained with the VQGAN loss for 40 epochs with a batch size of 128. We optimize using Adam with learning rate $1e-4$, $\beta_1 = 0.9$, and $\beta_1 = 0.95$. We apply an exponential moving average to the VQVAE weights with decay rate of 0.999. After the training is complete, our VQ-VAE model reached an rFID value of 2.20, which matches the rFID reported by Sun et al. (2024) on non-face-blurred ImageNet256.

**The baseline**    The baseline with a masked source distribution uses the cubic scheduler $\kappa_t = t^3$.

**The metric path.**    Our metric-induced probability path uses the euclidean distance of the token VQVAE embeddings as the distance function with lp being a free parameter:

$$\text{d}(x, x_1) = |\text{emb}(x) - \text{emb}(x_1)|_2^{\text{lp}}. \tag{110}$$

Furthermore, we parameterize $\beta_t$ as

$$\beta_t = c \left( \frac{t}{1-t} \right)^a , \tag{111}$$

with $c$ and $a$ being free parameters.

These three parameters are costly to search, because each configuration requires a separate model to train. We tune these parameters visually by plotting the samples along the conditional path and looking for configurations that make use of the whole time interval [0,1]. We settled on $a = 0.9$, $c = 3$ and $\mathrm{lp} = 4$ (see Figure 4)

| $t = 0.0$ | $t = 0.125$ | $t = 0.25$ | $t = 0.375$ | $t = 0.5$ | $t = 0.625$ | $t = 0.75$ | $t = 0.875$ | $t = 1.0$ |

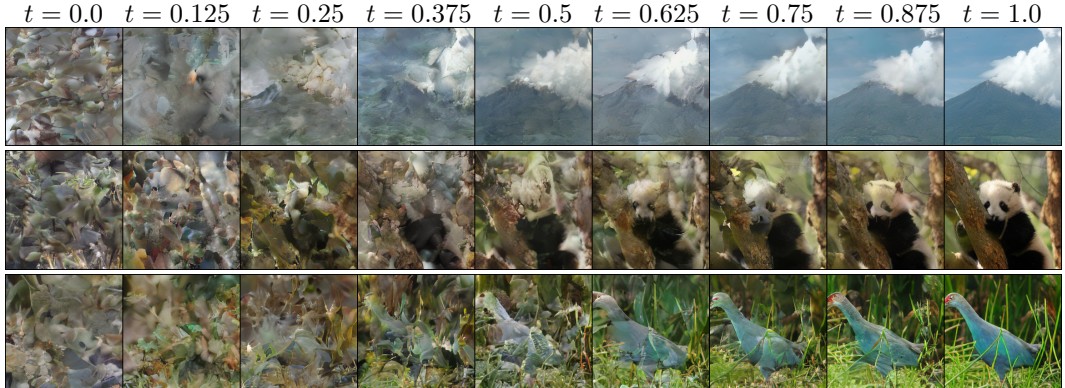

Figure 4: The conditional path for $a = 0.9$, $c = 3$ and $\mathrm{lp} = 4$. This path is advantageous because the the path smoothly interpolates from noise to image while utilizing the whole interval $t \in [0, 1]$.

**Latent Generative model details.** Our generative model uses the Llama architecture that is also used by the LlamaGen model (Sun et al., 2024). Our comparisons are done on the Llama-B architecture variant with 111M parameters. For training hyperparameters, we used the exact configuration proposed in Sun et al. (2024): batch size of 256, learning rate of 1e-4 with 2500 warmup steps, weight decay of 0.05, Adam optimizer with $\beta_1 = 0.9$ and $\beta_2 = 0.95$, gradient norm of 1.0 and class drop probability of 0.1. We used the same ten-crop data augmentation for training that (Sun et al., 2024) used.

Following the guidance of (Sun et al., 2024), the autoregressive and masked models were trained for 300 epochs. We found that the metric path model benefited from further training, so we trained this variant for 600 epochs.

The DFM models required minor architecture adjustments:

- The masked configuration uses non-causal attention.
- The metric path configuration uses non-causal attention and we also prepend a time embedding token (sinusoidal embedding) before the class label token to enable the model to learn the time dependency.

**Evaluation.** We report the FID of 50,000 generated images w.r.t. the training set. Note that our LlamaGen reproduction obtains a lower FID value then reported in Sun et al. (2024) (4.81 vs 5.46). This difference is due to us using the face-blurred variant of ImageNet. While Sun et al. (2024) compares against the pre-computed statistics of non-face-blurred ImageNet, we compile the statistics of face-blurred ImageNet, including training data augmentations.

**Ablations.** We show ablations for CFG scale (Table 5) and NFE (Table 6).

## F    RELATION TO SEDD (LOU ET AL., 2024)

In this section we explain the relation between our method and SEDD (Lou et al., 2024). We focus on three main points:

| CFG scale | 1.0 | 1.1 | 1.2 | 1.3 | 1.4 | 1.5 | 1.6 | 1.7 | 1.8 | 1.9 | 2.0 | 2.1 | 2.2 | 2.3 | 2.4 | 2.5 |
|---|---|---|---|---|---|---|---|---|---|---|---|---|---|---|---|---|
| LlamaGen, FID: | – | – | – | – | – | 5.77 | 5.18 | 4.91 | 4.81 | 5.02 | 5.26 | 5.63 | 6.11 | 6.56 | 7.12 | 7.70 |
| DFM masked path, NFE=100, FID: | 17.78 | 12.85 | 9.53 | 7.45 | 6.28 | 5.78 | 5.76 | 6.03 | 6.56 | 7.20 | 7.97 | – | – | – | – | – |
| DFM metric path, NFE=100, FID: | 8.58 | 5.99 | 4.87 | 4.50 | 4.82 | 5.47 | – | – | – | – | – | – | – | – | – | – |

Table 5: Ablation of CFG scale for LlamaGen and DFM models. The missing cells were not evaluated because they are far from the optima.

| NFE | 50 | 100 | 150 | 200 | 250 |
|---|---|---|---|---|---|
| DFM masked path, CFG=1.6, FID: | 5.73 | 5.72 | 5.74 | 5.71 | 5.82 |
| DFM metric path, CFG=1.3, FID: | 4.78 | 4.50 | 4.69 | 4.87 | 4.98 |

Table 6: Ablation of NFE for the DFM models.

1. *Generality of probability paths.* SEDD starting point is a diffusion matrix $Q_t^i(x^i, z^i)$ and requires a closed-form conditional probability $p_t(x^i|x_1^i)$ path solving the Kolmogorov equation (linear ODE) with this rate matrix. This entails solving a (general) $|\mathcal{T}|$ dimensional ODE which can be hard to do in closed form. Therefore SEDD resorts to rates of the form $Q_t^i(x^i, z^i) = \sigma_t Q^i(x^i, z^i)$. In contrast, our method offers a closed form generating rates (velocity) for *every* conditional probability path, see equation 16 and 26.

2. *Score-velocity conversion.* The concrete score function is a particular way to parameterize a probability velocity which is given by

$$u_t^i(x^i, z) = Q_t^i(x^i, z^i)s_t^i(x^i, z). \tag{112}$$

3. *Loss.* The training loss of SEDD can be seen as instance of our ELBO (36) when using the concrete score parameterization.

**Probability velocity vs. concrete score.** Using our notation, the noising process of SEDD taking a distribution $p_1$ at time $t = 1$, to a some simple distribution $p_0$ at time $t = 0$ is defined by the transition probability

$$\mathbb{P}(X_{t-h} = x \mid X_t = z) = \delta_z(x) + hQ_t(x, z) + o(h), \tag{113}$$

where $Q_t \in \mathbb{R}^{|\mathcal{S}| \times |\mathcal{S}|}$ is called *diffusion matrix* and it satisfies the rate conditions as in equation 5. The reverse process, taking the distribution $p_0$ at time $t = 0$ to the distribution $p_1$ at $t = 1$ is given by the diffusion matrix,

$$\bar{Q}_t(x, z) = Q_t(z, x)\frac{p_t(x)}{p_t(z)} \tag{114}$$

where the marginal $p_t$ is determined by the noising process (113) and $p_1$ the distribution at the boundary $t = 1$. The transition probability of the reverse process is

$$\mathbb{P}(X_{t+h} = x \mid X_t = z) = \delta_z(x) + h\bar{Q}_t(x, z) + o(h), \tag{115}$$

To make the process tractable, the noising diffusion matrix that is chosen only allows transitions from states $z \in \mathcal{S}$ to $x \in \mathcal{S}$ that differ by single token as in equation 6,

$$Q_t(x, z) = \sum_{i=1}^{D} Q_t^i(x^i, z^i) \prod_{j \neq i} \delta_{z^j}(x^j), \tag{116}$$

where $Q^i \in \mathbb{R}^{|\mathcal{T}| \times |\mathcal{T}|}$ and satisfy the rate conditions (5). In this case the diffusion matrix of the reverse process is,

$$\bar{Q}_t(x, z) = Q_t(z, x)\frac{p_t(x)}{p_t(z)} \tag{117}$$

$$= \sum_{i=1}^{D} Q_t^i(z^i, x^i) \prod_{j \neq i} \delta_{x^j}(z^j)\frac{p_t(x)}{p_t(z)} \tag{118}$$

$$= \sum_{i=1}^{D} Q_t^i(z^i, x^i)s_t^i(x^i, z) \prod_{j \neq i} \delta_{z^j}(x^j), \tag{119}$$

where $s_t(x^i, z)$ is called the *concrete score function* and it is defined as

$$s_t^i(x^i, z) = \frac{p_t(z^1, ..., z^{i-1}, x^i, z^{i+1}, ..., z^D)}{p_t(z^1, ..., z^{i-1}, z^i, z^{i+1}, ..., z^D)}. \tag{120}$$

Considering the boundary condition at time $t = 1$ to be the data distribution, $p_1 \equiv q$, since in our notation the velocity of reverse process is $u_t(x, z) = \bar{Q}_t(x, z)$ we have that (comparing equation 119 and equation 7)

$$u_t^i(x^i, z) = Q_t^i(z^i, x^i) s_t^i(x^i, z). \tag{121}$$

In the next paragraph we show that for the boundary condition $p_1 \equiv \delta_{x_1}$, the time marginal of the noising process is factorized,

$$p_t(x|x_1) = \prod_{i=1}^{D} p_t(x^i|x_1^i). \tag{122}$$

In this case the conversion from concrete score to the probability velocity is,

$$u_t(x^i, z^i|x_1^i) = Q_t(z^i, x^i) \frac{p_t(x^i|x_1^i)}{p_t(z^i|x_1^i)}. \tag{123}$$

Considering equation 9, we see that the relation between the concrete score and the probability velocity in equation 121 holds only if $Q_t^i(x^i, z^i)$ is independent of $x_1$.

**The conditional probability path.** The conditional probability path is the marginal of the noising process when taking $p_1 \equiv \delta_{x_1}$. Hence, the relation between the diffusion matrix $Q_t$ and the conditional probability path is given by an ODE,

$$\frac{d}{dt} p_{1-t}(x|x_1) = \sum_{z \in \mathcal{S}} Q_{1-t}(x, z) p_{1-t}(z|x_1) \tag{124}$$

$$= \sum_{z \in \mathcal{S}} \sum_{i=1}^{D} Q_{1-t}^i(x^i, z^i) \prod_{j \neq i} \delta_{z^j}(x^j) p_{1-t}(x_1). \tag{125}$$

One can check that indeed the factorized conditional probability path, *i.e.*, $p_t(x|x_1) = \prod_{i=1}^{D} p_t(x^i|x_1^i)$, is the (unique) solution to the above ODE in case that

$$\frac{d}{dt} p_{1-t}(x^i|x_1^i) = \sum_{z^i \in \mathcal{T}} Q_{1-t}^i(x^i, z^i) p_{1-t}(z^i|x_1^i). \tag{126}$$

The ODE in equation 126 is still too hard to solve in the general case, and some extra assumptions are in order if we hope to solve this equation in analytically. SEDD suggests the standard extra assumption that

$$Q_t^i(x^i, z^i) = \sigma_t Q^i(x^i, z^i), \tag{127}$$

where $\sigma : [0, 1] \to \mathbb{R}$, and $Q^i$ is constant in time. In this case the solution to equation 126 is

$$p_{1-t}(x^i|x_1^i) = \exp\left[\left(\int_0^t \sigma_s ds\right) Q^i\right](x^i, x_1^i). \tag{128}$$

The assumption in (127) significantly restricts the space of conditional probability paths.

In contrast, our point of view is arguably simpler: We start with an *arbitrary* conditional $p_t(x^i|x_1^i)$ and develop a closed-form expression for its generating velocity using equations (16) and (26).

For example, the generating process using our metric path as in equation 27 should be comparable to the reverse process given by some diffusion matrix,

$$Q_t^i(z^i, x^i) \frac{p_t(x^i|x_1^i)}{p_t(z^i|x_1^i)} = \bar{Q}_t^i(x^i, z^i|x_1^i) = u_t^i(x^i, z^i|x_1^i) = p_t(x^i|x_1^i) \dot{\beta}_t [d(z^i, x_1^i) - d(x^i, x_1^i)]_+, \tag{129}$$

assuming the diffusion matrix $Q_t(x^i, z^i)$ is restricted to Equation (127) we have that

$$Q^i(z^i, x^i) = \frac{p_t(z^i|x_1^i)}{\sigma_t} \dot{\beta}_t [d(z^i, x_1^i) - d(x^i, x_1^i)]_+ \tag{130}$$

on leading to a contradiction since the L.H.S is constant in time.

**SEDD training loss.** We derive the ELBO train loss for concrete score function as suggested in Lou et al. (2024) from our ELBO (36). To instantiate our ELBO we need to consider two reverse processes. The first correspond to the noising process (113) with the boundary condition $p_1 \equiv \delta_{x_1}$,

$$u_t(x^i, z^i | x_1^i) = \sigma_t Q^i(z^i, x^i) \frac{p_t(x^i | x_1^i)}{p_t(z^i | x_1^i)}. \tag{131}$$

The second correspond to the noising process (113) with the boundary condition $p_1 \equiv q$ (i.e., data distribution),

$$u_t^i(x^i, z) = \sigma_t Q^i(z^i, x^i) s_t^i(x^i, z). \tag{132}$$

Now we substitute the velocities in the ELBO (36),

$$\log p_1(x_1) \geq \int_0^1 \mathbb{E}_{x_t \sim p_t(\cdot | x_1)} \sum_{i=1}^{D} \sum_{y^i \neq x_t^i} \left[ u_t^i(y^i, x_t^i | x_1^i) - u_t^i(y^i, x_t) \right. \tag{133}$$

$$\left. + u_t^i(y^i, x_t^i | x_1^i) \log \left( \frac{u_t^i(y^i, x_t)}{u_t^i(y^i, x_t^i | x_1^i)} \right) \right] \mathrm{d}t \tag{134}$$

$$= \int_0^1 \mathbb{E}_{x_t \sim p_t(\cdot | x_1)} \sum_{i=1}^{D} \sum_{y^i \neq x_t^i} \sigma_t Q^i(x_t^i, y^i) \left[ \frac{p_t(y^i | x_1^i)}{p_t(x_t^i | x_1^i)} - s_t^i(y^i | x_t) \right. \tag{135}$$

$$\left. + \frac{p_t(y^i | x_1^i)}{p_t(x_t^i | x_1^i)} \log \left( \frac{p_t(x_t^i | x_1^i)}{p_t(y^i | x_1^i)} s_t^i(y^i | x_t) \right) \right] \mathrm{d}t \tag{136}$$

$$= \int_0^1 \mathbb{E}_{x_t \sim p_t(\cdot | x_1)} \sum_{i=1}^{D} \sum_{y^i \neq x_t^i} \sigma_t Q^i(x_t^i, y^i) \left[ - s_t^i(y^i | x_t) \right. \tag{137}$$

$$\left. + \frac{p_t(y^i | x_1^i)}{p_t(x_t^i | x_1^i)} \log \left( s_t^i(y^i | x_t) \right) - g \left( \frac{p_t(y^i | x_1^i)}{p_t(x_t^i | x_1^i)} \right) \right] \mathrm{d}t, \tag{138}$$

where $g(s) = s(\log(s) - 1)$.

# G    ADDITIONAL TABLES AND FIGURES

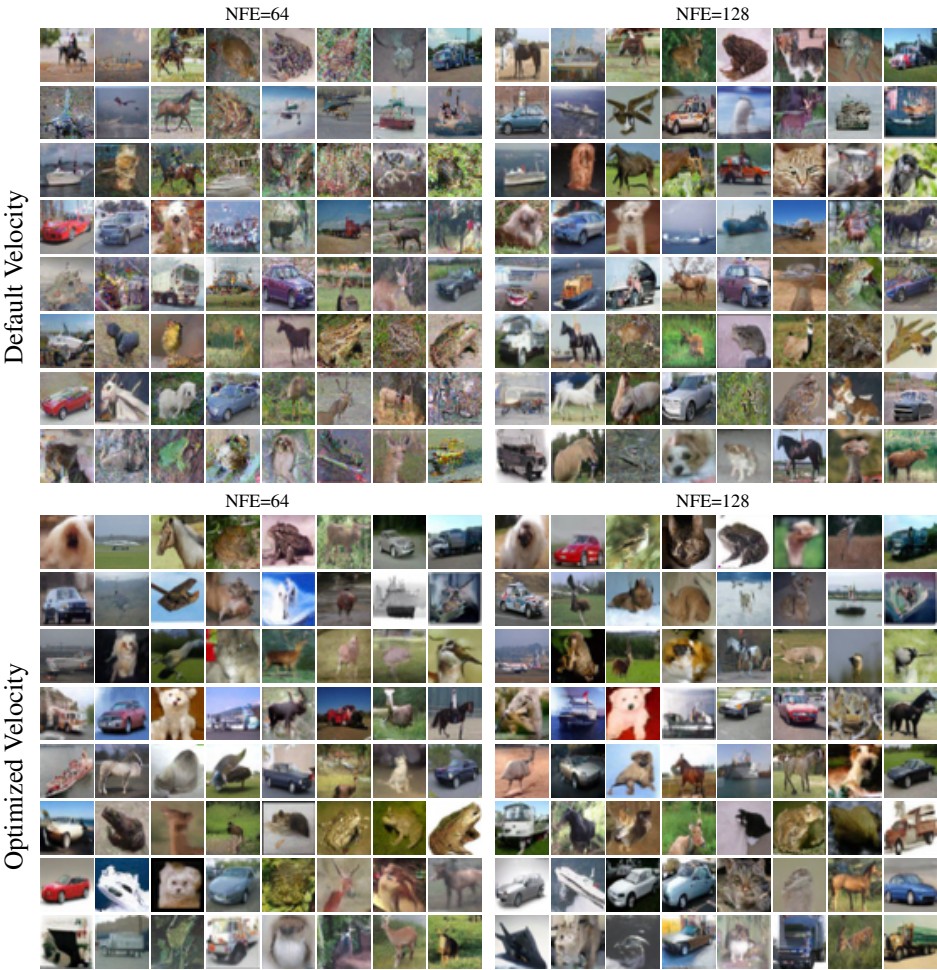

Figure 5: CIFAR10 Samples for 64 and 128 NFE, default velocities vs. optimized velocities. The default velocity we use is the velocity resulting from (26). The optimized velocity searches over (26) or (80), and also searches over the probability-preserving velocity (35) with varying weights. For each $8 \times 8$ table, same seed was used to generate the images.

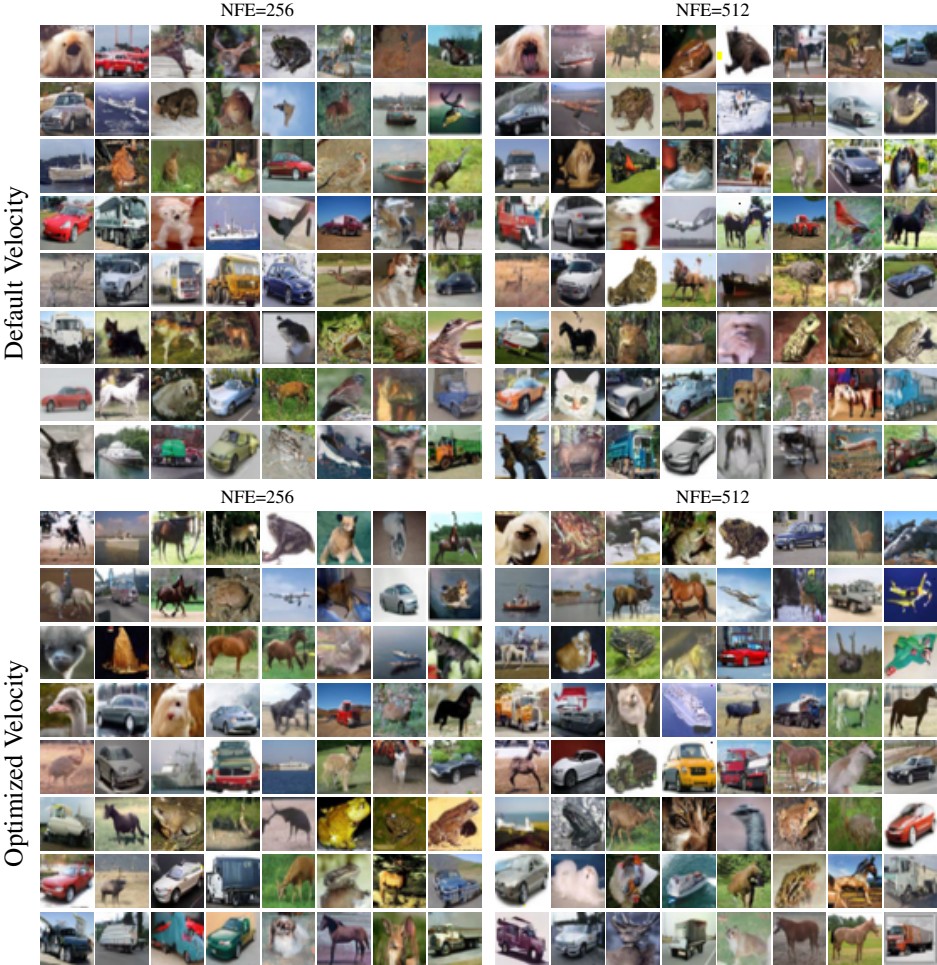

Figure 6: CIFAR10 Samples for 256 and 512 NFE, default velocities vs. optimized velocities. The default velocity we use is the velocity resulting from (26). The optimized velocity searches over (26) or (80), and also searches over the probability-preserving velocity (35) with varying weights. For each $8 \times 8$ table, same seed was used to generate the images.

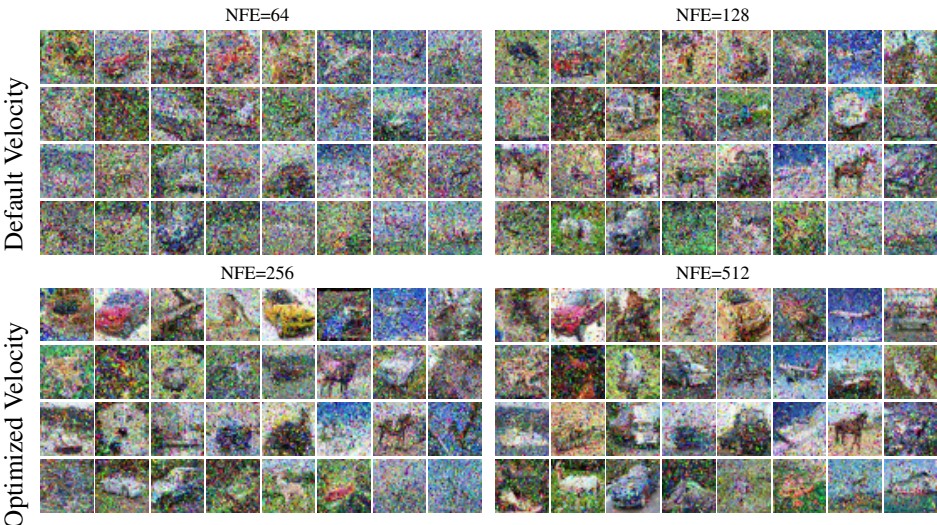

Figure 7: CIFAR10 samples generated from our model using the velocity from Campbell et al. (2024), which does not work for general probability paths such as our metric-induced paths. This is the same $p_{1|t}$ model as was used to generate samples for Figure 5 and Figure 6.

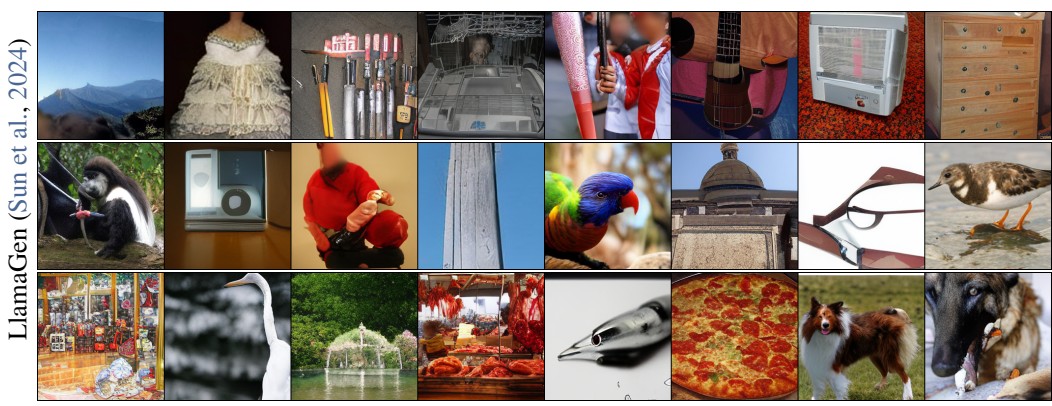

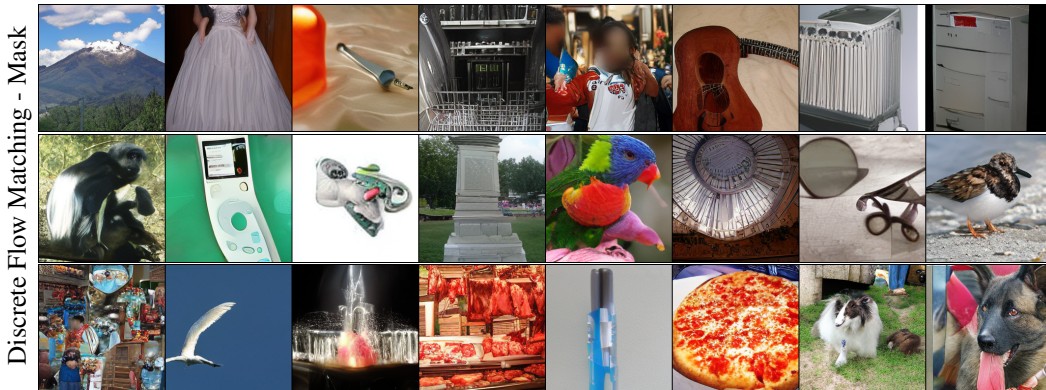

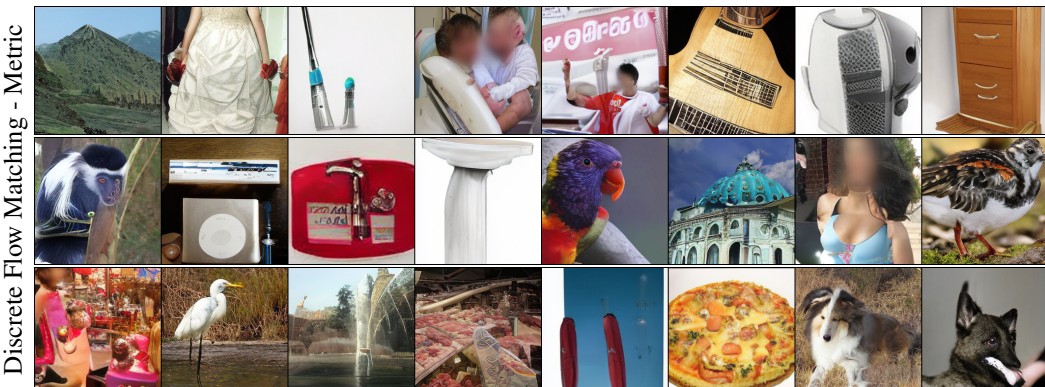

Figure 8: Non-curated generated samples for ImageNet256×256.

