# OpenReview forum: "Flow Matching with General Discrete Paths: A Kinetic-Optimal Perspective"
_ICLR.cc/2025/Conference — ICLR 2025 Oral_

### Official Review · Reviewer_vZy2 · 2024-10-31

**Soundness:** 3
**Presentation:** 3
**Contribution:** 3
**Rating:** 6
**Confidence:** 3

**Summary:**

This paper proposes extending discrete flow matching paths from a kinetic optimal perspective. Previous research in discrete flow matching has primarily focused on masking-related corruption paths. In contrast, this paper demonstrates that the design space can be broadened to include arbitrary probability paths. The authors explore two types of velocities: probability-advancing and probability-preserving. They also derive a closed-form velocity for the probability-advancing type and the ELBO for general discrete flow matching models.

**Strengths:**

1. This paper extends the commonly used probability paths in discrete diffusion to a more general class of paths, providing a closed-form velocity and ELBO. This approach has the potential to inspire further research and improve a broad range of discrete flow matching models.

2. The proposed method is general and can recover previously proposed methods, such as Gat et al. (2024).

3. The experiments are comprehensive, covering image, text, and material generation. Results show that the proposed path achieves better performance compared to previous mixture paths.

**Weaknesses:**

I have some concerns regarding the experimental results:

1. The results on ImageNet 256 fall short of state-of-the-art and even popular baselines, including both the autoregressive baseline and the proposed method. The network structure follows MaskGit; however, MaskGit reports an FID of 6, which is only half of the number reported in Table 3. What accounts for this discrepancy?

2. Why did the authors train the VQ-VAE for ImageNet rather than using off-the-shelf pretrained weights?

**Questions:**

Please see the weakness part.

---

> ### Author Response · Authors · 2024-11-26
>
> > The results on ImageNet 256 fall short of state-of-the-art and even popular baselines, including both the autoregressive baseline and the proposed method. The network structure follows MaskGit; however, MaskGit reports an FID of 6, which is only half of the number reported in Table 3. What accounts for this discrepancy?
>
> We thank the reviewer for raising this point. We have further investigated the gap in FID between our trained models and previous works [1] and identified the underlying causes:
> 1. Sub-optimal data preprocessing.
> 2. Sub-optimal architecture, which also included one dimensional positional encoding as opposed to two dimensional rotary encodings.
>
> To mitigate these differences in the revision we follow the setup of [1] for data preprocessing, architecture of both the VQVAE and the model, and training hyper-parameters, for exact details see updated Appendix E.4 in the revision. After retraining our model and baselines we are able to achieve SOTA results on tokenized ImageNet 256, see Table 3 in the revision.
>
>
> > Why did the authors train the VQ-VAE for ImageNet rather than using off-the-shelf pretrained weights?
>
> We trained our own VQVAE because we used the face-blurred variant of the ImageNet dataset in our experiment. As outlined in the paper introducing this variant (https://www.image-net.org/face-obfuscation/), the face-blurred variant has enhanced privacy. Note that our VQVAE reached the same rFID on face-blurred ImageNet as reported by Sun et. al. on unblurred ImageNet (rFID=2.20). We believe the differences between the datasets are rather minor and that we have properly reproduced prior results.
>
> [1] Sun, Peize, et al. "Autoregressive Model Beats Diffusion: Llama for Scalable Image Generation." arXiv preprint arXiv:2406.06525 (2024).
>
> [2] Chang, Huiwen, et al. "Maskgit: Masked generative image transformer." Proceedings of the IEEE/CVF Conference on Computer Vision and Pattern Recognition. 2022.

---

### Official Review · Reviewer_kVBR · 2024-10-31

**Soundness:** 4
**Presentation:** 4
**Contribution:** 4
**Rating:** 10
**Confidence:** 2

**Summary:**

This paper takes a holistic approach on discrete generative models. It pioneers the use of arbitrary discrete probability paths. The methods are validated on text generation, inorganic material generation and image generation.

**Strengths:**

1. This paper is very well written and succeeds in explaining concisely yet thoroughly the theory and background. It is well positioned in existing literature.

2. The authors propose holistic, novel methodologies for flow matching in discrete spaces.

3. The methods are experimentally validated on a broad set of modalities.

**Weaknesses:**

None.

**Questions:**

Is the use of the concept *flux* in this context novel, or already described in literature? If so, please provide a citation.

## Minor remarks
- There is a weird hyperlink at line 77 from the superscript 1 in $x^1$ to an equation in the appendix. This is also present in equation 6.
- line 174 'abusing a bit *the* previous ..'
- lines 202 - 206: it would be nice if the black triangles where vertically aligned

---

> ### Author Response · Authors · 2024-11-26
>
> > Is the use of the concept flux in this context novel, or already described in literature? If so, please provide a citation.
>
> The concept of flux is quite prevalent in physics and dynamical systems. In the context of discrete generative models, it was previously used in [1], where also the relation to flux in the context of continuous flows is explained in figure 2 of [1]. However, we are the first to formulate a kinetic energy optimization problem for the probability velocity as in equation 18. Considering the flux as the problem’s unknown leads to a convex problem which we are able to solve in closed form for certain cases (i.e., symmetric weights).
>
> > line 174 'abusing a bit the previous ..'
>
> We thank the reviewer for his comment, we fixed the typo.
>
> > lines 202 - 206: it would be nice if the black triangles where vertically aligned
>
> We thank the reviewer for his comment, we fixed the alignment of the triangles in Eq. (18).
>
> [1] Gat, Itai, et al. "Discrete flow matching." arXiv preprint arXiv:2407.15595 (2024).

---

> > ### Comment · Reviewer_kVBR · 2024-11-27
> >
> > I acknowledge your response, thanks.

---

### Official Review · Reviewer_S7CB · 2024-11-03

**Soundness:** 4
**Presentation:** 4
**Contribution:** 4
**Rating:** 8
**Confidence:** 2

**Summary:**

This paper proposes to broaden the design space for constructing better discrete flow generative models. To this end, the authors first decouple the probability paths from the velocity, effectively allowing to choose the velocity independently of the target probability path beyond the traditional masking method. Through a kinetic-optimal perspective, they propose closed-form velocities for a certain subset of symmetric kinetic energy, and further show that the mixture path model from previous work is kinetic-optimal when used with a specific scheduler. Additionally, a new tractable ELBO for mixture paths is introduced, which generalizes previous formulations. Experiments across modalities such as text, materials, and image generation demonstrate that the proposed framework outperforms existing methods.

**Strengths:**

The generality of the proposed design space is definitely a significant contribution of the paper. Indeed, the new framework supports finding symmetric kinetic-optimal velocities for arbitrary discrete probability paths, expanding design flexibility over traditional masking.

The paper has a strong theoretical foundation, and lays the ground for a more flexible approach to discrete generative modeling, which improves over state of the art, as highlighted by the experiments.

**Weaknesses:**

While the kinetic energy framework (Eq. 18) allows injecting problem dependent weighting $w_t(x,z)$ in theory, solving for arbitrary weights requires numerical approximation in practice, which can be challenging as mentioned by the authors. This can limit the practical potential of the design space.

**Questions:**

I would be interested to know how do DFM approaches compare with FM approaches in the current state of the art. For example, with equal training time and computational resources, how does discrete flow matching compare with continuous flow matching in the task of pixel space image generation?

---

> ### Author Response · Authors · 2024-11-26
>
> > While the kinetic energy framework (Eq. 18) allows injecting problem dependent weighting $w_t(x,z)$ in theory, solving for arbitrary weights requires numerical approximation in practice, which can be challenging as mentioned by the authors. This can limit the practical potential of the design space.
>
> We agree; we have opened up the opportunity for many future works to explore this direction by proposing a new framework for constructing velocities given probability paths. We note that it is possible to enable a larger set of weights by using efficient numerical solvers, but we decided to focus the current study on analytical solutions.
>
> We note that while the design space of the kinetic optimal velocity is limited, the design space of the probability path is not. Even having a single velocity formulation that can work with general discrete paths is already very useful, and converting to any other velocity for the same probability path can theoretically be done using the probability-preserving component proposed in Section 5 so the limitation is minimal.
>
>
> > I would be interested to know how do DFM approaches compare with FM approaches in the current state of the art. For example, with equal training time and computational resources, how does discrete flow matching compare with continuous flow matching in the task of pixel space image generation?
>
> Following the reviewer question we train a continuous flow matching (FM)~[1] CIFAR10 model with the Cond-OT scheduler. Similar to our discrete CIFAR10 model, we use the U-Net architecture as in [2]. This results in an FID of 2.59, which outperforms our discrete metric-induced path model. We mainly focus our paper on comparison to discrete models, and we note that staying within discrete space provides additional benefits such as neural compression. In this paper we considerably extended the design space of discrete flows, and showcase that it allows us to improve discrete flows on that task of image generation. Closing the gap to continuous flows is an important future goal, and we believe our work is a significant step toward achieving it. Furthermore, note that we have now also reproduced and outperformed LlamaGen on ImageNet256, a state-of-the-art discrete model when scaled up [3].
>
>
>
> [1] Lipman, Yaron, et al. "Flow matching for generative modeling." arXiv preprint arXiv:2210.02747 (2022).
>
> [2] Dhariwal, Prafulla, and Alexander Nichol. "Diffusion models beat gans on image synthesis." Advances in neural information processing systems 34 (2021): 8780-8794.
>
> [3] Sun, Peize, et al. "Autoregressive Model Beats Diffusion: Llama for Scalable Image Generation." arXiv preprint arXiv:2406.06525 (2024).

---

> > ### Comment · Reviewer_S7CB · 2024-12-02
> >
> > I would like to thank the authors for the clarification and the extra experiment on CIFAR10 comparing the proposed method with continuous flow matching. Naturally, I wasn't expecting the discrete model to outperform the continuous one, but as the authors point out, the proposed method contributes to closing that gap.
> >
> > With that, I thank the authors for their reply and I will keep my score.
> >
> > Best,

---

### Official Review · Reviewer_ZXqN · 2024-11-03

**Soundness:** 3
**Presentation:** 3
**Contribution:** 3
**Rating:** 6
**Confidence:** 3

**Summary:**

Most existing methods rely on masking for the corruption process, which leaves the design space of discrete diffusion and flows underexplored. This paper introduces a new design space by optimizing symmetric kinetic energy, allowing the proposed velocities to apply across multiple probability paths. Thus, the proposed discrete flow matching method enables desired probability paths for domain-specific applications. The effectiveness of this approach is validated across various modalities — including text generation, material generation, and image generation — where it outperforms mask-based methods, especially in language tasks, and achieves a new state-of-the-art in crystalline material generation.

**Strengths:**

1. The proposed method, Discrete Flow Matching (DFM), is theoretically well-justified through the use of kinetic optimal velocities and probability paths.

2. Extensive evaluations on text, crystalline material, and image generation tasks demonstrate the effectiveness of the proposed velocity formulation, achieving state-of-the-art results in material generation.

3. The construction of an infinite number of velocities for any chosen probability path broadens the design space for probability paths, enhancing applications for downstream tasks.

**Weaknesses:**

1. Typo in Line 50: “misleadingly actually” --> "actually misleading"

2. The velocity $u_t(x,z)$ is only defined for $x\neq z$ in Eq. (5).  The authors need to explicitly define $u_t(x,z)$ for the case where $x=z$, or explain why this case does not need to be defined separately.

3. What is $\bar{i}$ in Line 117? Define this when it is first introduced in the paper.

4. Line 227: Explicitly explain the relationship between the symmetric condition mentioned and the detailed balance equation in Markov Chains, if any. This would help clarify the theoretical foundations of the proposed method.

5. The authors are requested to provide a brief comparison of the key mathematical or conceptual differences between their kinetic optimal velocity formulation and the approach used in SEDD [1].

6. Typo in Line 511: “can be provide” --> "can provide"

7. Typo in Line 528: “We additional show generated” --> “We additionally show generated”

Overall, this paper is well-written with sound theoretical justification and experimental results supporting its main claims. The reviewer is unfamiliar with some related work (e.g. crystalline material generation) and is open to discussion with other reviewers/authors for a more comprehensive assessment.

**Questions:**

Please see the weaknesses section above.

---

> ### Author Response · Authors · 2024-11-26
>
> > The velocity $u_t(x,z)$ is only defined for $x \ne z$ in Eq. (5). The authors need to explicitly define $u_t(x,z)$ for the case where $x=z$, or explain why this case does not need to be defined separately.
>
> The probability velocity is first defined in equations 4 and 5 where we note that the probability velocity must satisfy two constraints called _Rate Conditions_:
> 1. For any $x \ne z$, $u_t(x,z) \ge 0$.
> 2. For any $z$, $\sum_x u_t(x,z)=0$.
>
> Hence, for any probability velocity $u_t(z,z)=-\sum_{x\ne z}u_t(x,z)$. This uniquely sets $u_t(z,z)$. We have made this clear also in the revision.
>
>
>
> > What is $\bar{i}$ in Line 117? Define this when it is first introduced in the paper.
>
> We thank the reviewer for this comment, indeed we did not define the $\bar{i}$ notation properly. We now removed this notation and provided an explicit expression instead (see equation 7 in the revision).
>
>
> > Line 227: Explicitly explain the relationship between the symmetric condition mentioned and the detailed balance equation in Markov Chains, if any. This would help clarify the theoretical foundations of the proposed method.
>
> We impose this condition in order to relax the problem statement (19), specifically, this condition allows us to drop the non-negative flux condition (19c) and arrive at a closed-form solution later on. Detailed balance usually shows up in deriving stationary processes. Here, we derive velocities specifically for non-stationary processes. It looks graphically similar to detailed balance but they are not related.
>
>
>
> > The authors are requested to provide a brief comparison of the key mathematical or conceptual differences between their kinetic optimal velocity formulation and the approach used in SEDD [1].
>
> Following this request we now added Appendix F in the revised paper, where we outline in detail the relations between our method and SEDD [1]. We agree this is important to clarify the landscape of discrete diffusion methods and thank the reviewer for this suggestion!
>
> In particular, we expand upon: (i) the fact that our work allows using a strict superset of the  probability paths possible in SEDD, introducing new and important design options for discrete diffusion/flow; (ii)  the exact conversion between our velocity and SEDD score when this conversion exists; and (iii) how their loss function is related to ours. We have further updated our experiments where we re-implement SEDD for apples-to-apples comparison; please see general response for details.
>
>
> [1] Lou, Aaron, Chenlin Meng, and Stefano Ermon. "Discrete diffusion language modeling by estimating the ratios of the data distribution." arXiv preprint arXiv:2310.16834 (2023).

---

> > ### Comment · Reviewer_ZXqN · 2024-12-02
> > **Official Comment by Reviewer ZXqN**
> >
> > Dear Authors,
> >
> > Thank you for adding the new results and discussion in the main text as well as in Appendix F. Given the similarities and generalization of the proposed approach over the prior work SEDD, I will keep my positive score regarding the paper.
> >
> > Best,
> >
> > Reviewer ZXqN

---

### Official Review · Reviewer_zfHf · 2024-11-04

**Soundness:** 3
**Presentation:** 3
**Contribution:** 3
**Rating:** 8
**Confidence:** 2

**Summary:**

This paper broadens discrete generative modeling by introducing discrete Flow Matching models with flexible, iterative probability paths using continuous-time Markov chains (CTMC). Unlike conventional masked models, this approach enables optimized, domain-specific probability paths.
Key contributions:
A decomposition of path velocities, making generation simpler.
Closed-form velocities that allow any probability path, unifying and expanding prior methods.
A kinetic energy-based optimization of paths, leading to new, source-dependent schedulers.
A general ELBO that improves performance on mixture paths, surpassing masked models.

**Strengths:**

I generally like this paper; it establishes a new framework for discrete flow matching and provides a method for constructing kinetic-optimal probability paths. The experiments across three downstream applications effectively demonstrate its real-world utility.

**Weaknesses:**

1. Is the construction in Section 4.2 unique, and is it general enough?
2. How does the proposed method relate to discrete rectified flow, and how do their equilibrium probability paths differ when using the same source and target domains?

**Questions:**

Could you provide more background on the mixture path and explain how it connects to the claim in Section 4.2?

---

> ### Author Response · Authors · 2024-11-26
>
> > Is the construction in Section 4.2 unique, and is it general enough?
>
> Yes, the construction in 4.2 is indeed unique: it is equivalent to the shortest path problem on the hyper-sphere. Note however, that 4.2 is constructed only for the weight choice as in equation 25. It is general in the sense it can be used to map any source distribution $p$ to any target $q$.
>
>
>
> > How does the proposed method relate to discrete rectified flow, and how do their equilibrium probability paths differ when using the same source and target domains?
>
> We are not sure whether the reviewer's intention by “discrete rectified flow” is related to Discrete Flow Matching[1] or Rectified Flow[2,3] on a discrete target data (e.g., text), hence we will consider both interpretations.
>
> **Discrete flow matching**
>
> Discrete Flow Matching [1] focuses on conditional probability path called the _mixture path_, of the form
>
> $$p_t(x^i|x_1^i)=(1-\kappa_t)p(x^i) + \kappa_t\delta_{x_1^i} (x^i). $$
>
> In contrast, we considerably extend the design space of discrete flow matching to allow **general** conditional probability paths. For example, we introduce a metric induced probability path in equation 27,
>
> $$p_t(x^i| x_1^i) =  \text{softmax}\left( -\beta_t d(x^i, x_1^i) \right),$$
>
> where $\beta_t$ is a monotonic scheduler and $d(x^i,x_1^i)$ is an arbitrary dissimilarity measure. We demonstrate the advantage of this extended family of probability paths in our Image generation experiments on CIFAR10 in Figure 2. and on ImageNet 256 in Table 3.
>
> Another example is given in equation 2 utilizing a token dependent scheduler,
>
> $$p_t(x^i|x_1^i)=(1-\kappa_t(x_1^i))p(x^i) + \kappa_t(x_1^i)\delta_{x_1^i} (x^i),$$
>
> and later in section 4.2 we show it is kinetic optimal for a certain weight of the kinetic energy.
>
> **Rectified flow for discrete data**
>
> There exists a work called Language Rectified flow~[3], where the discrete data is embedded in a continuous space (e.g., $\mathbb{R}^D$) and a latent rectified flow model is defined in the continuous latent space. They do not define a Markov process that directly acts in discrete space.
>
>
>
>
> > Could you provide more background on the mixture path and explain how it connects to the claim in Section 4.2?
>
> The mixture path is a particular choice of a conditional probability path that interpolates between source and target distributions using convex linear combinations, as introduced in section 2 equation 2,
>
> $$p_t(x^i|x_1^i)=(1-\kappa_t(x_1^i))p(x^i) + \kappa_t(x_1^i)\delta_{x_1^i} (x^i).$$
>
> Special cases of the mixture path were used in previous works [1,4,5,6,7], however no additional theoretical justification was given for it so far.
>
> In section 4.2 we show that the mixture path minimizes the kinetic energy with the specific weighting $w_t(x,z)=\frac{1}{p_t(x)}$.
>
>
> [1] Gat, Itai, et al. "Discrete flow matching." arXiv preprint arXiv:2407.15595 (2024).
>
> [2] Liu, Xingchao, Chengyue Gong, and Qiang Liu. "Flow straight and fast: Learning to generate and transfer data with rectified flow." arXiv preprint arXiv:2209.03003 (2022).
>
> [3] Zhang, Shujian, et al. "Language rectified flow: Advancing diffusion language generation with probabilistic flows." arXiv preprint arXiv:2403.16995 (2024).
>
> [4] Lou, Aaron, Chenlin Meng, and Stefano Ermon. "Discrete diffusion language modeling by estimating the ratios of the data distribution." arXiv preprint arXiv:2310.16834 (2023).
>
> [5] Campbell, Andrew, et al. "Generative flows on discrete state-spaces: Enabling multimodal flows with applications to protein co-design." arXiv preprint arXiv:2402.04997 (2024).
>
> [6] Shi, Jiaxin, et al. "Simplified and Generalized Masked Diffusion for Discrete Data." arXiv preprint arXiv:2406.04329 (2024).
>
> [7] Sahoo, Subham Sekhar, et al. "Simple and Effective Masked Diffusion Language Models." arXiv preprint arXiv:2406.07524 (2024).

---

### Author Response · Authors · 2024-11-26

We thank all the reviewers for their insightful review. We summarize the changes we did in the revision and additional experiments we ran following the reviewers' suggestions.

**ImageNet** Following reviewer vZy2 concerns, we have investigated the gap in FID between our models and results reported by previous works [1,2].  The main reason is: we used one-dimensional positional embedding, compared to [1,2], that used two-dimensional positional embeddings. Following this, we reproduced the results from LlamaGen [1] (which is a state-of-the-art discrete autoregressive model on imagenet256). We modified Table 3 accordingly, with state-of-the-art FID values. Note: the key finding remains unchanged, metric probability path outperforms both autoregressive modeling and masked construction.


**Text generation:** After the submission we realized that our ELBO computation was not consistent with previously reported methods and would result in slight differences in reported values. For this reason, unprompted by any reviewer, we have since updated our results for the text experiments. In particular, we reimplemented SEDD and MD4, hence, we ensured that the results do not differ due to implementation or architectural differences. Figure 1 has been updated with new values; our key finding remains: the kinetic optimal scheduler lies on the Pareto front more times than the linear scheduler and thus can be a strong default scheduler for future works. Table 1 has also been updated, and shows results trained on FineWeb, a recent high-quality larger (by an order of magnitude) dataset that many works are adopting for text generation, consistent with Figure 1. We will add an autoregressive baseline to Table 1. Note that our key findings remain: our proposed kinetical optimal choices outperform SEDD and MD4, which use masked construction with linear scheduler, on most zero-shot evaluations. Furthermore, we obtain better perplexity on the training set, and the use of non-mask source distributions can further drastically reduce generative perplexity.

**Relation to SEDD [3]:** In response to Reviewer ZXqN’s request, we now include more  details regarding the relation between our approach and SEDD. To summarize, we provide three key observations:

1. **Relation between score and velocity.** The relation between SEDD’s concrete score parameterization of the time-reversal rates and our velocity rate can be written in a closed-form, akin to the relation of score and velocity of Gaussian paths in the continuous case. We show we can always convert from a SEDD parameterization to equivalent velocities in our formulation, but not vice versa.

2. **Our probability paths strictly generalize SEDD’s paths.** SEDD builds probability paths by: (i) considering rate matrices of a forward (noising) process; (ii) figuring out its marginal probabilities in closed form; and (iii) expressing the reverse-process rate, which is used for sampling, using the forward-process rate and closed-form marginal probabilities. To accomplish (ii), SEDD requires solving an ODE in closed form and consequently introduces heavy restrictions upon the noising rate matrices, considerably limiting the reverse-processes that can be considered in SEDD. In contrast, our approach to build probability paths is: (i) writing down the desired probability path that interpolates the source and target; and (ii) find generating velocities for these probability paths. In particular we show that (ii) can be solved in closed form for general probability paths that form a strict superset of SEDD’s paths.

3. **SEDD loss is an instantiation of our ELBO.** The training loss of SEDD is a particular case of the ELBO in our formulation. This implies that SEDD and MD4, since they both use mask with a linear scheduler, differ mainly in the parameterization of the reverse-process.

Points 1 and 2 both impose restrictions on the probability paths / corruption processes that SEDD can work with. The derivation details and justifications for these claims are provided in Appendix F.

[1] Sun, Peize, et al. "Autoregressive Model Beats Diffusion: Llama for Scalable Image Generation." arXiv preprint arXiv:2406.06525 (2024).

[2] Chang, Huiwen, et al. "Maskgit: Masked generative image transformer." Proceedings of the IEEE/CVF Conference on Computer Vision and Pattern Recognition. 2022.

[3] Lou, Aaron, Chenlin Meng, and Stefano Ermon. "Discrete diffusion language modeling by estimating the ratios of the data distribution." arXiv preprint arXiv:2310.16834 (2023).

---

### Meta-Review · Area_Chair_FB8g · 2024-12-17

**Metareview:**

This paper makes a significant contribution to discrete generative modeling by broadening the design space of flow matching methods, allowing the use of arbitrary probability paths with a strong theoretical foundation grounded in kinetic-optimal velocities. Reviewers highlighted the paper's generality, elegant theoretical formulation, and its ability to unify and expand upon prior methods, while achieving state-of-the-art performance in tasks such as text and material generation. The experimental validation across diverse modalities, along with the clear and well-written presentation, further demonstrates the real-world utility and impact of the proposed approach.

**Additional Comments On Reviewer Discussion:**

During the rebuttal period, reviewers raised concerns regarding the experimental results on ImageNet, clarity in defining certain notations, and the relationship between the proposed method and prior works like SEDD. The authors addressed these by retraining models with improved data preprocessing and architecture to achieve state-of-the-art results, clarifying notation and theoretical justifications (e.g., velocity definitions and flux), and adding a detailed comparison with SEDD in Appendix F. Reviewers acknowledged the clarifications and new experiments, maintaining their positive assessments and highlighting the robustness and generality of the proposed framework.

---

### Decision · Program_Chairs · 2025-01-22

Accept (Oral)